# The Star Geometry of Critic-Based Regularizer Learning

**Oscar Leong**
Department of Statistics and Data Science
University of California, Los Angeles
oleong@stat.ucla.edu

**Eliza O'Reilly**
Department of Applied Mathematics and Statistics
Johns Hopkins University
eoreill2@jh.edu

**Yong Sheng Soh**
Department of Mathematics
National University of Singapore
matsys@nus.edu.sg

## Abstract

Variational regularization is a classical technique to solve statistical inference tasks and inverse problems, with modern data-driven approaches parameterizing regularizers via deep neural networks showcasing impressive empirical performance. Recent works along these lines learn task-dependent regularizers. This is done by integrating information about the measurements and ground-truth data in an unsupervised, critic-based loss function, where the regularizer attributes low values to likely data and high values to unlikely data. However, there is little theory about the structure of regularizers learned via this process and how it relates to the two data distributions. To make progress on this challenge, we initiate a study of optimizing critic-based loss functions to learn regularizers over a particular family of regularizers: gauges (or Minkowski functionals) of star-shaped bodies. This family contains regularizers that are commonly employed in practice and shares properties with regularizers parameterized by deep neural networks. We specifically investigate critic-based losses derived from variational representations of statistical distances between probability measures. By leveraging tools from star geometry and dual Brunn-Minkowski theory, we illustrate how these losses can be interpreted as dual mixed volumes that depend on the data distribution. This allows us to derive exact expressions for the optimal regularizer in certain cases. Finally, we identify which neural network architectures give rise to such star body gauges and when such regularizers have favorable properties for optimization. More broadly, this work highlights how the tools of star geometry can aid in understanding the geometry of unsupervised regularizer learning.

## 1   Introduction

The choice and design of regularization functionals to promote structure in data has a long history in statistical inference and inverse problems, with roots dating back to classical Tikhonov regularization [89]. In such problems, one is tasked with solving the following: given measurements $y \in \mathbb{R}^m$ of the form $y = \mathcal{A}(x_0) + \eta$ for some forward model $\mathcal{A} : \mathbb{R}^d \to \mathbb{R}^m$ and noise $\eta \in \mathbb{R}^m$, recover an estimate of the ground-truth signal $x_0 \in \mathbb{R}^d$. Typically, the challenge in such problems is that they are ill-posed, meaning that either there are no solutions, infinitely many solutions, or the problem is discontinuous in the data $y$ [7]. A pervasive and now classical technique is to identify regularization functionals $\mathcal{R} : \mathbb{R}^d \to \mathbb{R}$ such that, when minimized, promote structure that is present in the ground-

38th Conference on Neural Information Processing Systems (NeurIPS 2024).

truth signal. Well-known examples of hand-crafted regularizers to promote structure include the $\ell_1$-norm to promote sparsity [23, 16, 26, 88], total variation [74], the nuclear norm [29, 15, 70], and more generally, atomic norms [18, 12, 87, 78, 64].

With the development of modern machine learning techniques, we have seen a surge of data-driven methods to directly learn regularizers instead of designing regularizers in a hand-crafted fashion. Early works along these lines include the now-mature field of dictionary learning or sparse coding (see [60, 61, 84, 3, 6, 1, 2, 75, 76, 10, 81] and the surveys [56, 27] for more). More recently, we have seen powerful nonconvex regularizers parameterized by deep neural networks. These come in a variety of flavors, including those based on plug-and-play [91, 71], generative models [13, 39, 40, 8, 20], and regularization functionals directly parameterized by deep neural networks [53, 59, 80, 36, 47].

Despite the widespread success of such learning-based regularizers, several outstanding questions remain. In particular, it is unclear what type of structure is learned by such regularizers. For example, what is the relationship between the underlying data geometry and the type of regularizer found by data-driven approaches? Is the found regularizer "optimal" in some meaningful sense? If not, what properties of the data distribution are lost due to structural constraints placed on the regularizer?

In this work, we aim to tackle the above questions and others to further understand what types of regularizers are found via learning-based methods. We focus on a class of unsupervised methods that are *task-dependent*, i.e., those that learn a regularizer without paired training data, but still integrate information about the measurements in the learning process. This is done by identifying a loss inspired by variational representations of statistical distances, such as Integral Probability Metrics (IPMs) or divergences, where the regularizer plays the role of the "critic" or test function.

As an example, the recent adversarial regularization framework used in [53, 59, 80] learns a regularizer that assigns high "likelihood" to clean data $\mathcal{D}_r$ and low "likelihood" to noisy data $\mathcal{D}_n$ via a loss derived from a dual formulation of the 1-Wasserstein distance. This framework has showcased impressive empirical performance in learning data-driven regularizers, but there is still a lack of an overarching understanding of the structure of regularizers learned. Moreover, while this Wasserstein distance interpretation has shown to be useful, it is natural to consider if other losses can be derived from this "critic-based" perspective to learn regularizers.

## 1.1 Our contributions

In order to make progress on these challenges, we first fix a family of regularization functionals to analyze. We aim for this family to be (i) expressive (can describe both convex and nonconvex regularizers), (ii) exhibit properties akin to regularizers used in practice, and (iii) tractable to analyze. A family of functionals that satisfy such criteria are *gauges of star bodies*. In particular, we will consider regularization functionals of the form

$$\|x\|_K := \inf\{t > 0 : x \in tK\}$$

where $K$ is a *star body*, i.e., a compact subset of $\mathbb{R}^d$ with $0 \in \text{int}(K)$ such that for each $x \in \mathbb{R}^d \setminus \{0\}$, the ray $\{tx : t > 0\}$ intersects the boundary of $K$ exactly once. Such regularizers are nonconvex for general star bodies $K$, but note that $\|\cdot\|_K$ is convex if and only if $K$ is a convex body. Any norm is the gauge of a convex body, but this class also includes nonconvex quasinorms such as the $\ell_q$-quasinorm for $q \in (0, 1)$.

Focusing on the above class of functionals, we aim to understand the structure of a regularizer $\|\cdot\|_K$ found by solving an optimization problem of the form

$$\inf_{\|\cdot\|_K \in \mathcal{F}} \mathcal{H}\left(\|\cdot\|_K; \mathcal{D}_r, \mathcal{D}_n\right) \tag{1}$$

where $\mathcal{H}(\cdot; \mathcal{D}_r, \mathcal{D}_n) : \mathcal{F} \to \mathbb{R}$ compares the values $\|\cdot\|_K$ assigns to $\mathcal{D}_r$ and $\mathcal{D}_n$ and $\mathcal{F}$ is a class of functions on $\mathbb{R}^d$. The adversarial regularization framework of [53] is recovered by setting $\mathcal{H}(f; \mathcal{D}_r, \mathcal{D}_n) = \mathbb{E}_{\mathcal{D}_r}[f(x)] - \mathbb{E}_{\mathcal{D}_n}[f(x)]$ and $\mathcal{F} = \text{Lip}(1) := \{f : \mathbb{R}^d \to \mathbb{R} : f \text{ is 1-Lipschitz}\}$.

Our contributions are as follows:

1. Using tools from star geometry [41, 33] and dual Brunn-Minkowski theory [54], we prove that under certain conditions, the solution to (1) under the adversarial regularization frame-work of [53] can be exactly characterized. In particular, we show that the objective is

equivalent to a *dual mixed volume* between our star body $K$ and a data-dependent star body $L_{r,n}$. This allows us to exploit known bounds on the dual mixed volume via (dual) Brunn-Minkowski theory.

2. We investigate new critic-based loss functions $\mathcal{H}(\cdot; \mathcal{D}_r, \mathcal{D}_n)$ in (1) inspired by variational representations of divergences for probability measures. We specifically analyze $\alpha$-divergences and show how they can give rise to loss functionals with dual mixed volume interpretations. We also experimentally show that such losses can be competitive for learning regularizers in a simple denoising setting.

3. We conclude with results showcasing when these star body regularizers exhibit useful optimization properties, such as weak convexity, as well as analyzing what types of neural network architectures give rise to star body gauges.

This paper is organized as follows: Section 2 analyzes optimal adversarial regularizers using star geometry. We establish a general existence result in Section 2.1 and use dual mixed volumes to characterize the optimal star body regularizer under certain assumptions in Section 2.2. Visual examples are provided in Section 2.3. Section 3 introduces new critic-based losses for learning regularizers inspired by $\alpha$-divergences. An empirical comparison between neural network-based regularizers learned using these losses and the adversarial loss is presented in Section 3.1. Section 4 examines computational properties of star body regularizers, such as beneficial properties for optimization and relevant neural network architectures. We conclude with a discussion in Section 5.

## 1.2 Related work

We discuss the broader literature on learning-based regularization in Section A of the appendix and focus on optimal regularization here. To our knowledge, there are few works that analyze the optimality of a regularizer for a given dataset. The authors of the original adversarial regularization framework of [53] analyzed the case when $\mathcal{D}_r$ is supported on a manifold and $\mathcal{D}_n$ is related to $\mathcal{D}_r$ via the push-forward of the projection operator onto the manifold. They showed that in this case the distance function to the manifold is an optimal regularizer, but uniqueness does not hold (i.e., other regularizers could be optimal as well). Our work complements such results by analyzing the structure of the optimal regularizer for a variety of $\mathcal{D}_r$ and $\mathcal{D}_n$ which may not be related in this way. In [4] the authors analyze the optimal Tikhonov regularizer in the infinite-dimensional Hilbert space setting, and show that the optimal regularizer is independent of the forward model and only depends on the mean and covariance of the data distribution.

The two most closely related papers to ours are [90] and [52]. In [90], the authors aim to characterize the optimal convex regularizer for linear inverse problems. Several notions of optimality for convex regularizers are introduced (dubbed compliance measures) and the authors establish that canonical low-dimensional models, such as the $\ell_1$-norm for sparsity, are optimal for sparse recovery under such compliance measures. In [52], similarly to this work, the authors analyze the optimal regularizer amongst the family of star body gauges for a given dataset. They also leverage dual Brunn-Minkowski theory to show that the optimal regularizer is induced by a data-dependent star body, whose radial function depends on the density of the data distribution. Interestingly, these results also characterize data distributions for which convex regularization is optimal and they provide several examples. The present work is novel in relation to [52] for several reasons. First, our work introduces novel theory and a new framework for understanding critic-based regularizer learning, addressing a significant gap in existing theoretical foundations for unsupervised regularizer learning. This new setting also brings novel technical challenges that were not present in [52]. For example, the losses in Section 3 exhibit more complicated dependencies on the star body of interest, leading to the need for new analysis. Such losses are also experimentally analyzed in Section C.1. Finally, we present new results that are relevant to downstream applications of star body regularizers in Section 4.

## 1.3 Notation

In this section, we provide some brief background on star geometry and define notation used in the main body of the paper. For further details, please see Section B.1 in the appendix and the sources [77, 33, 41] for more. We say that a closed set $K \subseteq \mathbb{R}^d$ is *star-shaped* (with respect to the origin) if for all $x \in K$, we have $[0, x] \subseteq K$ where, for two points $x, y \in \mathbb{R}^d$, we define the line segment $[x, y] := \{(1 - t)x + ty : t \in [0, 1]\}$. $K$ is a *star body* if it is a compact star-shaped

set such that for every $x \neq 0$, the ray $R_x := \{tx : t > 0\}$ intersects the boundary of $K$ exactly once. Equivalently, $K$ is a star body if its *radial function* $\rho_K$ is positive and continuous over the unit sphere $\mathbb{S}^{d-1}$, where $\rho_K$ is defined as $\rho_K(x) := \sup\{t > 0 : t \cdot x \in K\}$. It follows that the gauge function of $K$ satisfies $\|x\|_K = 1/\rho_K(x)$ for all $x \in \mathbb{R}^d$ such that $x \neq 0$. Let $\mathcal{S}^d$ to be the space of all star bodies on $\mathbb{R}^d$. For $K, L \in \mathcal{S}^d$, it is easy to see that $K \subseteq L$ if and only if $\rho_K \leqslant \rho_L$. The *kernel* of a star body $K$ is the set of all points for which $K$ is star-shaped with respect to, i.e., $\ker(K) := \{x \in K : [x,y] \subseteq K, \forall y \in K\}$. Note that $\ker(K)$ is a convex subset of $K$ and $\ker(K) = K$ if and only if $K$ is convex. For a parameter $\gamma > 0$, we define the following subset of $\mathcal{S}^d$ consisting of *well-conditioned* star bodies with nondegenerate kernels: $\mathcal{S}^d(\gamma) := \{K \in \mathcal{S}^d : \gamma B^d \subseteq \ker(K)\}$ where $B^d := \{x \in \mathbb{R}^d : \|x\|_{\ell_2} \leqslant 1\}$. For two distributions $P$ and $Q$, let $P \circledast Q$ denote their convolution, i.e., $P \circledast Q$ is the distribution of $X + Y$ where $X \sim P$ and $Y \sim Q$. Finally, for a function $f$ and a measure $P$, let $f_\#(P)$ denote the push-forward measure of $P$ under $f$, i.e., for all measurable subsets $A$, we have $f_\#(P)(A) := P(f^{-1}(A))$.

## 2 Adversarial star body regularization

To illustrate our tools and results, we first consider regularizer learning under the adversarial regularization framework of [53]. Given two distributions $\mathcal{D}_r$ and $\mathcal{D}_n$ on $\mathbb{R}^d$ and setting $\mathcal{H}(f; \mathcal{D}_r, \mathcal{D}_n) = \mathbb{E}_{\mathcal{D}_r}[f(x)] - \mathbb{E}_{\mathcal{D}_n}[f(x)]$ and $\mathcal{F} := \text{Lip}(1)$ in (1), we aim to understand minimization of the functional $F(K; \mathcal{D}_r, \mathcal{D}_n) := \mathbb{E}_{\mathcal{D}_r}[\|x\|_K] - \mathbb{E}_{\mathcal{D}_n}[\|x\|_K]$ over all star bodies $K$ such that $x \mapsto \|x\|_K$ is 1-Lipschitz. A result due to [73] shows that $\|\cdot\|_K$ is 1-Lipschitz if and only if the unit ball $B^d \subseteq \ker(K)$. Here and throughout this paper, it will be useful to think of $\mathcal{D}_r$ as the distribution of ground-truth, clean data while $\mathcal{D}_n$ is a user-defined distribution describing noisy, undesired data. Using the notation from Section 1.3, we are interested in analyzing

$$\inf_{K \in \mathcal{S}^d(1)} F(K; \mathcal{D}_r, \mathcal{D}_n). \tag{2}$$

### 2.1 Existence of minimizers

As the problem (2) requires minimizing a functional over a structured subset of the infinite-dimensional space of star bodies, even basic questions such as the existence of minimizers are unclear. Despite this, one can exploit tools in the star geometry literature to obtain guarantees regarding this problem. The proof of this result exploits Lipschitz continuity of the objective functional and local compactness properties of $\mathcal{S}^d(\gamma)$, proved in [52], akin to the celebrated Blaschke's Selection Theorem from convex geometry [77]. We defer the proof to the appendix in Section B.2.

**Theorem 2.1.** *For any two distributions $\mathcal{D}_r$ and $\mathcal{D}_n$ on $\mathbb{R}^d$, we have that*

$$F(K; \mathcal{D}_r, \mathcal{D}_n) \geqslant W_1(\mathcal{D}_r, \mathcal{D}_n) \text{ for all } K \in \mathcal{S}^d(1)$$

*where $W_1(\cdot, \cdot)$ is the 1-Wasserstein distance between two distributions. Moreover, if $\mathbb{E}_{\mathcal{D}_i}[\|x\|_{\ell_2}] < \infty$ for each $i = r, n$, then we always have that minimizers exist:*

$$\arg\min_{K \in \mathcal{S}^d(1)} F(K; \mathcal{D}_r, \mathcal{D}_n) \neq \emptyset.$$

### 2.2 Minimization via dual Brunn-Minkowski theory

We now aim to understand the structure of minimizers to the above problem. To do this, we first show that the objective in (2) can be interpreted as the dual mixed volume between $K$ and a data-dependent star body. To begin, we start with a definition.

**Definition 2.2** (Definition 2* in [54]). Given two star bodies $K, L \in \mathcal{S}^d$, the $i$-th dual mixed volume between $L$ and $K$ for $i \in \mathbb{R}$ is given by

$$\tilde{V}_i(L, K) := \frac{1}{d} \int_{\mathbb{S}^{d-1}} \rho_L(u)^{d-i} \rho_K(u)^i \mathrm{d}u.$$

One can think of dual mixed volumes as functionals that measure the size of a star body $K$ relative to another star body $L$. Note that for all $i$, $\tilde{V}_i(K, K) = \frac{1}{d} \int_{\mathbb{S}^{d-1}} \rho_K(u)^d \mathrm{d}u = \text{vol}_d(K)$ is the usual $d$-dimensional volume of $K$. Of particular interest to us will be the case $i = -1$.

To see how our main objective can be interpreted as a dual mixed volume, we need to be able to summarize our distributions $\mathcal{D}_r$ and $\mathcal{D}_n$ in such a way that can naturally be related to the gauge $\|\cdot\|_K$. Since the gauge is positively homogenous, it is characterized by its behavior on the sphere $\mathbb{S}^{d-1}$. Hence, given a distribution, we require a "summary statistic" that describes the distribution in each unit direction $u \in \mathbb{S}^{d-1}$. Since star bodies are precisely defined by the distance of the origin to their boundary in each unit direction, we ideally would like to summarize the distribution in a similar fashion. For a distribution $\mathcal{D}$ with density $p$ on $\mathbb{R}^d$, define the map

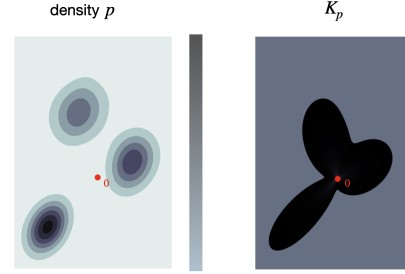

$$\rho_p(u) := \left( \int_0^\infty t^d p(tu) \mathrm{d}t \right)^{1/(d+1)}, \ u \in \mathbb{S}^{d-1}. \quad (3)$$

Figure 1: (Left) Contours of the Gaussian mixture model density $p$. (Right) The star body $K_p$ induced by the radial function (3).

This function measures the average mass the distribution accrues in each unit direction and how far this mass lies from the origin. More precisely, one can show [52] that $\rho_p^{d+1}$ is the density of the measure $\mu_{\mathcal{D}}(\cdot) := \mathbb{E}_{\mathcal{D}}\left[ \|x\|_{\ell_2} \mathbf{1}_{\{x/\|x\|_{\ell_2} \in \cdot\}} \right]$. If the map (3) is positive and continuous over the unit sphere, it defines a unique data-dependent star body. In Figure 1, we give an illustrative example showing how the geometry of the data distribution relates to the data-dependent star body. Here, the data distribution is given by a Gaussian mixture model with 3 Gaussians.

We now aim to understand the structure of minimizers of the functional $K \mapsto F(K; \mathcal{D}_r, \mathcal{D}_n)$. As we will show in our main result, the map (3) aids in interpreting the functional via a dual mixed volume. Extrema of such dual mixed volumes have been the object of study in convex and star geometry for some time. We cite a seminal result due to Lutwak.

**Theorem 2.3** (Special case of Theorem 2 in [54]). *For star bodies $K, L \in \mathcal{S}^d$, we have*

$$\tilde{V}_{-1}(L, K)^d \geqslant \mathrm{vol}_d(L)^{d+1} \mathrm{vol}_d(K)^{-1},$$

*and equality holds if and only if $K$ and $L$ are dilates, i.e., there exists a $\lambda > 0$ such that $K = \lambda L$.*

Armed with this inequality, we show that under certain conditions, we can guarantee the existence and uniqueness of minimizers of the above map. Note that optimal solutions are always defined up to scaling. To remove this degree of freedom, we impose an additional constraint on the volume of the solution. We chose a unit-volume constraint for simplicity, but changing the constraint would simply scale the optimal solution.

**Theorem 2.4.** *Suppose $\mathcal{D}_r$ and $\mathcal{D}_n$ are distributions on $\mathbb{R}^d$ that are absolutely continuous with respect to the Lebesgue measure with densities $p_r$ and $p_n$, respectively. Suppose $\rho_{p_r}$ and $\rho_{p_n}$ as defined in (3) are continuous over the unit sphere and that $\rho_{p_r}(u) > \rho_{p_n}(u) \geqslant 0$ for all $u \in \mathbb{S}^{d-1}$. Then, there exists a star body $L_{r,n} \in \mathcal{S}^d$ such that the unique solution to the problem*

$$\min_{K \in \mathcal{S}^d : \mathrm{vol}_d(K) = 1} \mathbb{E}_{\mathcal{D}_r}[\|x\|_K] - \mathbb{E}_{\mathcal{D}_n}[\|x\|_K]$$

*is given by $K_* := \mathrm{vol}_d(L_{r,n})^{-1/d} L_{r,n}$. If $B^d \subseteq \ker(K_*)$, then $K_*$ is, in fact, the unique solution to (2) with the additional constraint $\mathrm{vol}_d(K) = 1$.*

*Proof Sketch.* One can prove that for each $i = r, n$, $\mathbb{E}_{\mathcal{D}_i}[\|x\|_K] = \int_{\mathbb{S}^{d-1}} \rho_{p_i}(u)^{d+1} \rho_K(u) \mathrm{d}u$. Then since $\rho_{p_r} > \rho_{p_n} \geqslant 0$ on the unit sphere, we have that the map $\rho_{r,n}(u) := (\rho_{p_r}(u)^{d+1} - \rho_{p_n}(u)^{d+1})^{1/(d+1)}$ is positive and continuous on $\mathbb{S}^{d-1}$. We can then prove that it is the radial function of a new data-dependent star body $L_{r,n}$. Combining the previous two observations yields $\mathbb{E}_{\mathcal{D}_r}[\|x\|_K] - \mathbb{E}_{\mathcal{D}_n}[\|x\|_K] = d\tilde{V}_{-1}(L_{r,n}, K)$. Applying Theorem 2.3 obtains the final result. We defer the full proof to Section B.2. $\square$

*Remark* 2.5 (Uniqueness guarantees). We highlight that in our result we are able to obtain uniqueness of the solution everywhere. This is notably different than previous guarantees in the Optimal Transport literature, where the optimal transport potential for the 1-Wasserstein loss is not unique, but can be

shown to be unique $\mathcal{D}_n$-almost everywhere [58]. A significant reason for this is that our optimization problem is over the family of star body gauges, which have additional structure over the general class of 1-Lipschitz functions. In particular, star bodies are uniquely defined by their radial functions and, hence, dual mixed volumes can exactly specify them (see [54] for more details).

*Remark* 2.6 (Distributional assumptions). Note that the conditions of this theorem require that the density-induced map $\rho_p(\cdot)$ must be a valid radial function (i.e., positive and continuous over the unit sphere). Many distributions satisfy these assumptions. For example, if the distribution's density is of the form $p(x) = \psi(\|x\|_L^q)$ where $\int_0^\infty t^d \psi(t^q) \mathrm{d}t < \infty$, $q > 0$, and $L \in \mathcal{S}^d$, then the map $\rho_p(\cdot)$ is a valid radial function. See [52] for more examples of distributions that satisfy these assumptions.

*Remark* 2.7 (Finite-data regime). While the conditions of this theorem would not apply to the case when $\mathcal{D}_r$ and $\mathcal{D}_n$ are empirical distributions, we prove in the appendix (Section D.1) that these results are stable in the sense that as the amount of available data goes to infinity, the solutions in the finite-data regime converge (up to a subsequence) to a population minimizer. The proof exploits tools from variational analysis, such as $\Gamma$-convergence [14].

*Remark* 2.8 (Implications for inverse problems). In the original framework of [53], the authors considered $\mathcal{D}_n = \mathcal{A}_\#^\dagger(\mathcal{D}_y)$ where $\mathcal{A}$ is a linear forward operator in an inverse problem. Since inverse problems are typically under-constrained, such a distribution would be singular and not satisfy the assumptions of the theorem. One can solve this, however, by convolving $\mathcal{D}_n$ with a Gaussian to ensure it has full measure.

*Remark* 2.9 (Scaling distributions). One can always reweight the objective so that the assumptions of the theorem are satisfied. This is due to positive homogeneity of the gauge, which guarantees $\mathbb{E}_\mathcal{D}[\|\lambda x\|_K] = \lambda \mathbb{E}_\mathcal{D}[\|x\|_K]$ for any distribution $\mathcal{D}$. In particular, if $\rho_{p_r} > \rho_{p_n}$ is not satisfied, one can choose a scaling $\lambda$ so that $\rho_{p_r} > \lambda^{1/(d+1)} \rho_{p_n}$. Inspecting the proof of Theorem 2.4, one can see that the result goes through when applied to the objective $\mathbb{E}_{\mathcal{D}_r}[\|x\|_K] - \lambda \mathbb{E}_{\mathcal{D}_n}[\|x\|_K]$.

## 2.3 Examples

We now provide examples of optimal regularizers via different choices of $\mathcal{D}_r$ and $\mathcal{D}_n$. Further examples can be found in the appendix in Section D.3. Throughout these examples, the star body induced by $\mathcal{D}_r$ and $\mathcal{D}_n$ is denoted by $L_r$ and $L_n$, respectively.

**Example 1** (Gibbs densities with $\ell_1$- and $\ell_2$-norm energies). *Suppose the distributions $\mathcal{D}_r$ and $\mathcal{D}_n^\alpha$ for some $\alpha \geqslant 1$ are given by Gibbs densities with $\ell_1$- and $\ell_2$-norm energies, respectively:*

$$p_r(x) = e^{-\|x\|_{\ell_1}} / (c_d \operatorname{vol}_d(B_{\ell_1})) \text{ and } p_n^\alpha(x) = e^{-\alpha\sqrt{d}\|x\|_{\ell_2}} / (c_d \operatorname{vol}_d(B^d/(\alpha\sqrt{d}))).$$

*Here $c_d := \Gamma(d+1)$ where $\Gamma(\cdot)$ is the usual Gamma function. The star bodies induced by $p_r$ and $p_n^\alpha$ are dilations of the $\ell_1$-ball and $\ell_2$-ball, respectively. Denote these star bodies by $L_r$ and $L_n^\alpha$, respectively. Then, $L_{r,n}^\alpha$ is defined by the difference of radial functions via*

$$\rho_{L_{r,n}^\alpha}(u) := \left( c_r \|x\|_{\ell_1}^{-(d+1)} - c_n (\alpha\sqrt{d}\|x\|_{\ell_2})^{-(d+1)} \right)^{1/(d+1)}$$

*where $c_r := \int_0^\infty t^d \exp(-t)/(c_d \operatorname{vol}_d(B_{\ell_1}))$ and $c_n := \int_0^\infty t^d \exp(-t)/(c_d \operatorname{vol}_d(B^d/(\alpha\sqrt{d})))$. We visualize the geometry of $L_{r,n}^\alpha$ for different values of $\alpha \geqslant 1$ in Figure 2 for $d = 2$. We see that for directions such that the boundaries of $L_r$ and $L_n^\alpha$ are far apart (namely, the $\pm e_i$ directions), the optimal regularizer assigns small values. This is because such directions are considered highly likely under the distribution $\mathcal{D}_r$ and less likely under $\mathcal{D}_n^\alpha$. However, for directions in which the boundaries are close (e.g., in the $[0.5, 0.5]$-direction), the regularizer assigns high values, since such directions are likely under the noise distribution $\mathcal{D}_n^\alpha$. This aligns with the aim of the objective function – namely, that a regularizer should assign low values to likely directions and high values to unlikely ones.*

**Example 2** (Toy inverse problem). *Consider the following example where $x \sim \mathcal{D}_r := \mathcal{N}(0, \Sigma)$ where $\Sigma \in \mathbb{R}^{d \times d}$ and we have measurements $y = Ax$ where $A \in \mathbb{R}^{m \times d}$ with rank $m \leqslant d$. We consider the case when $\mathcal{D}_n := A_\#^\dagger(\mathcal{D}_y) \circledast \mathcal{N}(0, \sigma^2 I_d)$ where $y \sim \mathcal{D}_y$ if and only if $y = Ax$ where $x \sim \mathcal{D}_r$. Let $d = 2$, $\Sigma = UU^T$, and $A = [1, 0] = e_1^T \in \mathbb{R}^{1 \times 2}$. Then $A^\dagger = e_1$. Note that $y \sim \mathcal{D}_y$ if and only if $y = e_1^T U z = u_1^T z$ where $u_1^T$ is the first row of $U$ and $z \sim \mathcal{N}(0, I_2)$. By standard properties of Gaussians, $\mathcal{D}_y = \mathcal{N}(0, \|u_1\|_{\ell_2}^2)$. Hence $\mathcal{D}_n = \mathcal{N}(0, D_\sigma)$ where $D_\sigma := \operatorname{diag}(\|u_1\|_{\ell_2}^2 + \sigma^2, \sigma^2) \in \mathbb{R}^{2 \times 2}$. We visualize this example in Figure 3. We see that the regularizer induced by $L_{r,n}$ penalizes directions in the row span of $A$, but does not for directions in the kernel of $A$.*

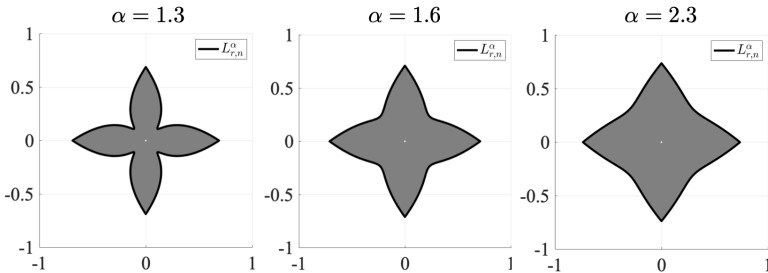

Figure 2: We plot $L_{r,n}^\alpha$ from Example 1 for different values of $\alpha$: (Left) $\alpha = 1.3$, (Middle) $\alpha = 1.6$, and (Right) $\alpha = 2.3$. A full figure with $L_r$ and $L_n^\alpha$ can be found in Section D.3.

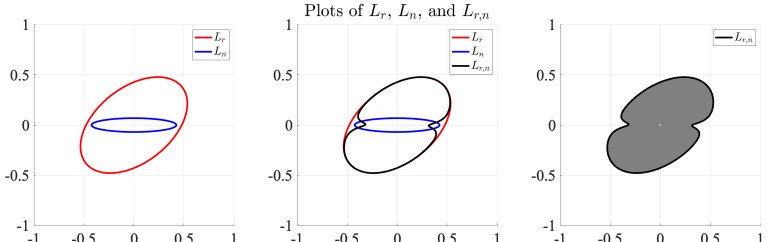

Figure 3: We visualize the distributions from Example 2 when $\Sigma = [0.5477, 0.2739; 0, 0.5477] \in \mathbb{R}^{2\times 2}$, $\sigma^2 = 0.01$, and we set $D_\sigma := 0.01 \operatorname{diag}(\|u_1\|_{\ell_2}^2 + \sigma^2, \sigma^2)$: (Left) the boundaries of $L_r$ and $L_n$, (Middle) the boundaries of $L_r$, $L_n$, and $L_{r,n}$ and (Right) $L_{r,n}$.

## 3 Critic-based loss functions via $f$-divergences

Inspired by the use of the variational representation of the Wasserstein distance to define loss functions to learn regularizers, we consider whether other divergences can give rise to valid loss functions and if they can be interpreted as dual mixed volumes. A general class of divergences of interest will be $f$-divergences: for a convex function $f : (0, \infty) \to \mathbb{R}$ with $f(1) = 0$, if $P \ll Q$,
$D_f(P||Q) := \int_{\mathbb{R}^d} f\left(\frac{\mathrm{d}P}{\mathrm{d}Q}\right) \mathrm{d}Q$.

$\alpha$-**Divergences:** For $\alpha \in (-\infty, 0) \cup (0, 1)$, there is a variational representation of the $\alpha$-divergence where $f = f_\alpha$ with $f_\alpha(x) = \frac{x^\alpha - \alpha x - (1-\alpha)}{\alpha(\alpha-1)}$ [69]:

$$D_{f_\alpha}(P||Q) := D_\alpha(P||Q) := \frac{1}{\alpha(1-\alpha)} - \inf_{h:\mathbb{R}^d \to (0,\infty)} \left( \mathbb{E}_Q\left[\frac{h(x)^\alpha}{\alpha}\right] + \mathbb{E}_P\left[\frac{h(x)^{\alpha-1}}{1-\alpha}\right] \right). \quad (4)$$

Note that the domain of functions in this variational representation consists of positive functions, which aligns with the class of gauges $\|\cdot\|_K$. Many well-known divergences are $\alpha$-divergences for specific $\alpha$, such as the $\chi^2$-divergence ($\alpha = -1$) and the (squared) Hellinger distance ($\alpha = 1/2$) [69]. We discuss in the appendix (Section B.3.1) how one can derive a general loss based on $\alpha$-divergences, but focus on a particular case here to illustrate ideas.

**The Hellinger Distance:** Setting $\alpha = 1/2$ and performing an additional change of variables in (4), we have the following useful representation of the (squared) Hellinger distance:

$$H^2(P||Q) := 2 - \inf_{h>0} \mathbb{E}_P[h(x)] + \mathbb{E}_Q[h(x)^{-1}].$$

This motivates minimizing the following functional, which carries a similar intuition to (2) that the regularizer should be small on real data and large on unlikely data:

$$K \mapsto \mathbb{E}_{\mathcal{D}_r}[\|x\|_K] + \mathbb{E}_{\mathcal{D}_n}[\|x\|_K^{-1}]. \quad (5)$$

The following theorem, proved in Section B.3, shows that this loss functional is equal to single dual mixed volume between a data-dependent star body $L_r$ and a star body that depends on $\mathcal{D}_r$, $\mathcal{D}_n$, and

$K$. We can show that the equality cases of the lower bound only hold for dilations within a specific interval, with only two specific star bodies achieving equality.

**Theorem 3.1.** *Suppose $\mathcal{D}_r$ and $\mathcal{D}_n$ admit densities $p_r$ and $p_n$, respectively, such that the following maps are positive and continuous over the unit sphere:*

$$u \mapsto \rho_{\mathcal{D}_r}(u)^{d+1} := \int_0^\infty t^d p_r(tu)\mathrm{d}t \text{ and } u \mapsto \tilde{\rho}_{\mathcal{D}_n}(u)^{d-1} := \int_0^\infty t^{d-2} p_n(tu)\mathrm{d}t.$$

*Let $L_r$ and $\tilde{L}_n$ denote the star body with radial function $\rho_{\mathcal{D}_r}$ and $\tilde{\rho}_{\mathcal{D}_n}$, respectively. Then, for any $K \in \mathcal{S}^d$, there exists a star body $K_{r,n}$ that depends on $\mathcal{D}_r, \mathcal{D}_n$, and $K$ such that the functional (5) is equal to $d\tilde{V}_{-1}(L_r, K_{r,n})$. We also have the inequality $\tilde{V}_{-1}(L_r, K_{r,n}) \geqslant \mathrm{vol}_d(K_{r,n})^{-1/d} \mathrm{vol}_d(L_r)^{(d+1)/d}$ with equality if and only if $K_{r,n}$ is a dilate of $L_r$. Moreover, there exists a $\lambda_* := \lambda_*(\mathcal{D}_r, \mathcal{D}_n) > 0$ such that for every $\lambda \in (0, \lambda_*]$, there are only two star bodies $K_{+,\lambda}$ and $K_{-,\lambda}$ that satisfy $K_{r,n} = \lambda L_r$:*

$$\rho_{K_{\pm,\lambda}}(u) := \frac{\rho_{L_r}(u)^d}{2\lambda \rho_{\tilde{L}_n}(u)^{d-1}} \left( 1 \pm \sqrt{1 - 4\lambda^2 \left( \frac{\rho_{\tilde{L}_n}(u)}{\rho_{L_r}(u)} \right)^{d-1}} \right)$$

*Remark* 3.2. Among the two star bodies $K_{+,\lambda}, K_{-,\lambda}$ that achieve $K_{r,n} = \lambda L_r$, we argue that $K_{+,\lambda}$ induces a better regularizer than $K_{-,\lambda}$. As seen in previous examples, we would like our regularizer $\| \cdot \|_K$ to assign low values to likely data and high values to unlikely data. Equivalently, we would like $\rho_K(\cdot)$ to be large on likely data and small on unlikely data. This can be stated in terms of the geometry of $L_r$ and $\tilde{L}_n$: we would like $\rho_K$ to be large when $\rho_{L_r}$ is large and $\rho_{\tilde{L}_n}$ is small. Likewise, $\rho_K$ should be small when $\rho_{\tilde{L}_n}$ is large and $\rho_{L_r}$ is small. Due to the small sign difference between $K_{+,\lambda}$ and $K_{-,\lambda}$, $K_{+,\lambda}$ better captures this intuition. We show a visual example of this in Figure 4, where $\mathcal{D}_r$ and $\mathcal{D}_n$ are the same distributions as in Example 1.

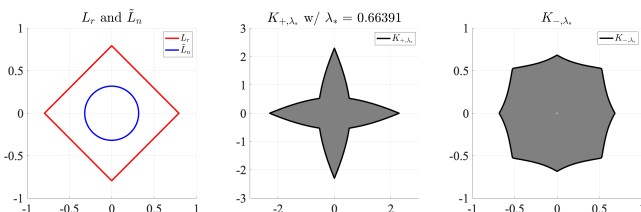

Figure 4: (Left) The star bodies $L_r$ and $\tilde{L}_n$ induced by the distributions $\mathcal{D}_r$ and $\mathcal{D}_n$ from from Theorem 3.1 and Example 1 with $\alpha = 0.5$. Then we have (Middle) $K_{+,\lambda_*}$ and (Right) $K_{-,\lambda_*}$ as defined in Theorem 3.1. Note that $K_{+,\lambda_*}$ better captures the geometry of a regularizer that assigns higher likelihood to likely data and lower likelihood to unlikely data, while $K_{-,\lambda_*}$ does not.

## 3.1 Empirical comparison with adversarial regularization

We now aim to understand whether the Hellinger-based loss (5) can provide a practical loss function to learn regularizers. To do this, we consider denoising on the MNIST dataset [50]. We take 10000 random samples from the MNIST training set (constituting our $\mathcal{D}_r$ distribution) and add Gaussian noise with variance $\sigma^2 = 0.05$ (constituting our $\mathcal{D}_n$ distribution). The goal is then to reconstruct test samples from the MNIST dataset corrupted with Gaussian noise of the same variance seen during training. The regularizers were parameterized via a deep convolutional neural network with positively homogenous Leaky ReLU activation functions, giving rise to a star-shaped regularizer. They were trained using the adversarial loss and Hellinger-based loss (5). We also used the gradient penalty term from [53] for both losses. Once each regularizer $\mathcal{R}_\theta$ has been trained, we then denoise a new noisy MNIST digit $y$ by minimizing $x \mapsto \|x - y\|_{\ell_2}^2 + \lambda \mathcal{R}_\theta(x)$ via gradient descent.

As a baseline, we compare both methods to Total Variation (TV) regularization [44]. We note that in [53], it was recommended to set $\lambda := 2\tilde{\lambda}$ where $\tilde{\lambda} := \mathbb{E}_{\mathcal{N}(0,\sigma^2 I)}[\|z\|_{\ell_2}]$, which was motivated by the fact that the regularizer will be 1-Lipschitz. We found our Hellinger-based regularizer performed best with $\lambda := 5.1\tilde{\lambda}^2$. We thus also compare to the adversarial regularizer with a further tuned parameter $\lambda$. Please see Section C.1 of the appendix for implementation details and a further discussion of regularization parameter choices.

We show the average Peak Signal-to-Noise Ratio (PSNR) and mean squared error (MSE) over 100 test images in the table below. We see that the Hellinger-based loss gives competitive performance relative to the adversarial regularizer and TV regularization, while the tuned adversarial regularizer slightly outperforms all methods. These promising results suggests that these new $\alpha$-divergence based losses are potentially worth exploring from a practical perspective as well.

|                    | PSNR  | MSE    |
| ------------------ | ----- | ------ |
| Hellinger          | 23.06 | 0.0052 |
| Adversarial        | 22.32 | 0.0064 |
| Adversarial (tuned)| 24.14 | 0.0041 |
| TV                 | 20.52 | 0.0091 |

## 4  Computational considerations

We now discuss various issues relating to employing such star body regularizers. In particular, we discuss optimization-based concerns due to the (potential) nonconvexity of $x \mapsto \|x\|_K$ and how one can use neural networks to learn star body regularizers in high-dimensions.

**Weak convexity:**  When such a regularizer is employed in inverse problems, one may require minimizing a cost of the form $x \mapsto \|y - \mathcal{A}(x)\|_{\ell_2}^2 + \lambda\phi(\|x\|_K)$. Here $\phi(\cdot)$ is a continuous monotonically increasing function (e.g., $\phi(t) = t^q$ for $q > 0$). While the first term is convex for linear $\mathcal{A}(\cdot)$, the second term can be highly nonconvex for general star bodies. One could enforce searching for a convex body as opposed to a star body, but convexity is strictly more limited in terms of expressibility. There is a large body of recent work [24, 25, 19, 36, 80] that has analyzed weak convexity in optimization, showing that while nonconvex, weakly convex functions can be provably optimized using simple first-order methods in some cases. Based on this, it is important to understand under what conditions is $\phi(\|\cdot\|_K)$ a weakly convex function. Recall that a function $f$ is $\rho$-weakly convex if $f(\cdot) + \frac{\rho}{2}\|\cdot\|_{\ell_2}^2$ is convex. For certain functions $\phi(\cdot)$, there is a natural way to determine if $\phi(\|\cdot\|_K)$ is weakly convex.

In particular, in considering the quantity $\|\cdot\|_K^2 + \frac{\rho}{2}\|\cdot\|_{\ell_2}^2$, there is a natural way in which one can describe this functional as the gauge of a new star body.

**Definition 4.1** ([55])**.** For two star bodies $K, L \in \mathcal{S}^d$ and scalars $\alpha, \beta \geqslant 0$ (both not zero), the harmonic $q$-combination for $q \geqslant 1$ is the star body $\alpha \diamond K \,\hat{+}_q\, \beta \diamond L$ whose radial function is defined by

$$\rho_{\alpha \diamond K \hat{+}_q \beta \diamond L}(u)^{-q} := \alpha\rho_K(u)^{-q} + \beta\rho_L(u)^{-q}.$$

Observe that the scaling operation $\alpha \diamond K := \{\alpha x : x \in K\}$ differs from the usual $\alpha K := \{\alpha x : x \in K\}$ since $\alpha\rho_K^{-q} = \rho_{\alpha^{-q}K}^{-q}$ so that $\alpha \diamond K = \alpha^{-q}K$.

Notice that in our context, we have a similar combination given by the harmonic 2-combination of $K$ and the unit ball $B^d$: $M_{2,\rho} := M_{2,\rho}(K) := K \,\hat{+}_2\, \frac{\rho}{2}B^d$. By the definition of the harmonic 2-combination, the gauge of $M_{2,\rho}$ satisfies $\|x\|_{M_{2,\rho}}^2 = \|x\|_K^2 + \frac{\rho}{2}\|x\|_{\ell_2}^2$. We thus have the following Proposition, proven in Section B.4, showing the equivalence between the weak convexity of $\|\cdot\|_K^2$ and the convexity of $M_{2,\rho}$. We also show a visual example in Section D.4 of a data-dependent star body whose gauge squared is weakly convex.

**Proposition 4.2.** *Let $K \in \mathcal{S}^d$. Then $x \mapsto \|x\|_K^2$ is $\rho$-weakly convex if and only if $M_{2,\rho}$ is convex.*

**Deep neural network-based parameterizations:**  Since deep neural networks allow for efficient learning in high-dimensions, it is natural to ask under what conditions does the regularizer $\mathcal{R}_\theta(x) := f_L^{\theta_L} \circ f_{L-1}^{\theta_{L-1}} \circ \cdots \circ f_1^{\theta_1}(x)$ define a star body regularizer, i.e., when is the set $K_\theta := \{x \in \mathbb{R}^d : \mathcal{R}_\theta(x) \leqslant 1\}$ a star body? As star bodies are in one-to-one correspondence with radial functions, one can answer this question by studying conditions under which the map $u \mapsto 1/\mathcal{R}_\theta(u)$ defines a radial function. This leads to the following Proposition, which is proven in Section B.4.

**Proposition 4.3.** *For $L \in \mathbb{N}$, consider a regularizer $\mathcal{R} : \mathbb{R}^d \to \mathbb{R}$ of the form $\mathcal{R}(x) := \mathcal{G} \circ f_L \circ \cdots \circ f_1(x)$ where $\mathcal{G} : \mathbb{R}^{d_L} \to \mathbb{R}$ and each $f_i : \mathbb{R}^{d_{i-1}} \to \mathbb{R}^{d_i}$ for $i \in [L]$ satisfy the following conditions:*

*(i)* $d = d_0 \leqslant d_1 \leqslant d_2 \leqslant \cdots \leqslant d_L$,

*(ii)* $\mathcal{G}(\cdot)$ *is non-negative, positively homogenous, continuous, and only vanishes at the origin,*

*(iii)* $f_i(\cdot)$ *is injective, continuous, and positively homogenous.*

*Then the set $K := \{x \in \mathbb{R}^d : \mathcal{R}(x) \leqslant 1\}$ is a star body in $\mathbb{R}^d$ with radial function $\rho_K(u) := 1/\mathcal{R}(u)$.*

Below we discuss the different types of commonly employed layers and activation functions that satisfy the conditions of the above Proposition.

**Example 3** (Feed-forward layers). *Any intermediate layer $f_i : \mathbb{R}^{d_{i-1}} \to \mathbb{R}^{d_i}$ can be parameterized by a single-layer feedforward MLP with no biases: $f_i(x) := \sigma_i(W_i x)$ where $W_i \in \mathbb{R}^{d_i} \times \mathbb{R}^{d_{i-1}}$ has rank $d_{i-1}$. For the activation function $\sigma_i(\cdot)$, one can ensure injectivity and positive homogeneity by choosing the Leaky ReLU activation $\mathrm{LeakyReLU}_\alpha(t) := \mathrm{ReLU}(t) + \alpha\mathrm{ReLU}(-t)$ where $\alpha \in (0,1)$. Observe that this map is positively homogenous, continuous, and bijective. The layer $f_i$ thus satisfies condition (iii). For the final layer $\mathcal{G}(\cdot)$, note that any norm on $\mathbb{R}^d$ satisfies condition (ii).*

**Example 4** (Residual layers). *If $d_i = d_{i-1}$, one could use residual layers of the form $f_i(x) := x + g_i(x)$ where $g_i : \mathbb{R}^{d_i} \to \mathbb{R}^{d_i}$ is a Lipschitz continuous, positively homogenous function with $\mathrm{Lip}(g_i) < 1$. This can be achieved by having $g_i$ be a neural network with 1-Lipschitz positively homogenous activations (such as $\mathrm{ReLU}$), no biases, and weight matrices with norm strictly less than 1. Then each $f_i$ is positively homogenous, continuous, and invertible [11], satisfying condition (iii).*

*Remark* 4.4 (Star-shaped regularizers). Note that if one does not require the set $K := \{x \in \mathbb{R}^d : \mathcal{R}(x) \leqslant 1\}$ to be a star body, but simply star-shaped, then there is more flexibility in the neural network architecture used. In particular, one simply needs the layers to be positively homogenous and continuous, and the output layer be non-negative, positively homogenous, and continuous. Invertibility would not be necessary, so that the dimensions $d_i$ need not be increasing. This can easily be achieved via linear convolutional layers and positively homogenous activation functions. In fact, in the original work [53], experiments were done with a network precisely of this form.

*Remark* 4.5 (Positive homogeneity of activations). While our theory for neural networks mainly applies to positively homogenous activation functions such as ReLU and LeakyReLU, we show in Section C.2 of the appendix that focusing on such activations does not limit performance. Specifically, we compare networks with non-positively homogenous activation functions (such as Tanh or the Gaussian Error Linear Unit (GELU)) with a network exclusively using Leaky ReLU activations in denoising. We show that the Leaky ReLU-based regularizer outperforms regularizers with non-positively homogenous activations. This potentially highlights empirical benefits of positive homogeneity, which warrants further analysis.

## 5  Discussion

In this work, we studied optimal task-dependent regularization over the class of regularizers given by star body gauges. Our analysis focused on learning approaches that optimize a critic-based loss function derived from variational representations of probability measures. Utilizing tools from dual Brunn-Minkowski theory and star geometry, we precisely characterized the optimal regularizer in specific cases and provided visual examples to illustrate our findings. Overall, our work underscores the utility of star geometry in enhancing our understanding of data-driven regularization.

**Limitations and future work:**  There are numerous exciting directions for future research. While our analysis concentrated on formulations based on the Wasserstein distance and $\alpha$-divergence, it would be valuable to extend the analysis to other critic-based losses for learning regularizers. Additionally, it would be interesting to give a precise characterization of global minimizers for the Hellinger-inspired loss (5). Relaxing the assumptions in our results is another significant challenge. For instance, Theorem 2.4 hinges on a "containment" property of the distributions, and finding ways to relax this assumption could broaden the applicability of our theory. Moreover, many of our results assume a certain regularity in the densities of our distributions. Addressing this would involve extending dual mixed volume inequalities to operate on star-shaped sets rather than star bodies. Finally, it is crucial to assess the performance of these regularizers in solving inverse problems. This includes studying questions regarding sample complexity, algorithmic properties, robustness to noise, variations in the forward model, and scenarios with limited clean data [22, 45, 32, 51, 85, 86, 21, 31].

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

# Contents

## A  Further discussion on related work

We survey topics relating to data-driven regularization and convex geometry for inverse problems. For a more in-depth discussion on learning-based regularization methods, we refer the reader to the following survey articles [7, 62].

**Regularization via generative modeling:** Generative networks have demonstrated their effectiveness as signal priors in various inverse problems, including compressed sensing [45, 13, 57],

denoising [42], phase retrieval [39, 40, 43, 79], and blind deconvolution [9]. Early approaches in this area leveraged "push-forward" generators such as Generative Adversarial Networks (GANs) [35] and Variational Autoencoders (VAEs) [46]. In some cases, generative models can serve as traditional regularizers by providing access to likelihoods, which can be incorporated as penalty functions into the objective [8]. Alternatively, they can constrain reconstructions to lie on the natural image manifold, ensuring that solutions are realistic and consistent with the underlying data distribution. More recently, there has been a surge of interest in techniques based on diffusion models [20, 30, 21, 72].

**Explicit regularization:** There have also been recent works that have directly learned regularizers as penalty functions in a data-driven fashion. Early works in this area include the now-mature field of dictionary learning (see the surveys [56, 27] for more). Regularization by Denoising [71] constructs an explicit regularization functional using an off-the-shelf deep neural network-based denoiser. The works [53, 59, 80] learn a regularizer via the critic-based adversarial loss inspired by the Wasserstein distance. The difference in these works lie in the way the neural network is parameterized to enforce certain properties, such as convexity or weak convexity. Recently, [28] developed an approach that can approximate the proximal operator of a potentially nonconvex data-driven regularizer. Other works along these lines include [36, 4, 37].

**Convex geometry:** Our work is also a part of a line of research showcasing the tools of convex and star geometry in machine learning. In statistical inference tasks and inverse problems, the utility of convex geometry cannot be understated. In particular, complexity measures of convex bodies, such their Gaussian width [18, 17] or statistical dimension [5], have proven useful in developing sample complexity guarantees for a variety of inverse problems [64, 66, 67, 48, 65, 83]. Additionally, convex geometry has been fruitful in problems such as function estimation from noisy data [38, 34, 63, 82, 49].

# B  Proofs

## B.1  Preliminaries on star and convex geometry

In this section, we provide relevant background concerning convex bodies and star bodies. For a more detailed treatment of each subject, we recommend [77] for convex geometry and [33, 41] for star geometry.

We say that a closed set $K \subseteq \mathbb{R}^d$ is *star-shaped* (with respect to the origin) if for all $x \in K$, the line segment $[0, x] := \{tx : t \in [0, 1]\} \subseteq K$. In order for $K$ to be considered a *star body*, we require $K$ to be a compact star-shaped set such that for every $x \neq 0$, the ray $R_x := \{tx : t > 0\}$ intersects the boundary of $K$ exactly once. Note that every convex body (a compact, convex set with the origin its interior) is a star body.

In an analogous fashion to the correspondence between convex bodies and their support functions, star bodies are in one-to-one correspondence with radial functions. For a star body $K$, its *radial function* is defined by

$$\rho_K(x) := \sup\{t > 0 : t \cdot x \in K\}.$$

It follows that the gauge function of $K$ satisfies $\|x\|_K = 1/\rho_K(x)$ for all $x \in \mathbb{R}^d$ such that $x \neq 0$. The radial function can also help capture important geometric information about $K$. For example, one can show that $\mathrm{vol}_d(K) = \frac{1}{d} \int_{\mathbb{S}^{d-1}} \rho_K^d(u) \mathrm{d}u$.

For $K, L \in \mathcal{S}^d$, it is easy to see that $K \subseteq L$ if and only if $\rho_K \leqslant \rho_L$. We say that $L$ is a *dilate* of $K$ if there exists $\lambda > 0$ such that $L = \lambda K$; that is, $\rho_L = \lambda \rho_K$. In addition, given a linear transformation $\phi \in GL(\mathbb{R}^d)$, one has $\rho_{\phi K}(x) = \rho_K(\phi^{-1}x)$ for all $x \in \mathbb{R}^d \setminus \{0\}$.

An additional important aspect of star bodies is their kernels. Specifically, we define the *kernel* of a star body $K$ as the set of all points for which $K$ is star-shaped with respect to, i.e., $\ker(K) := \{x \in K : [x, y] \subseteq K, \forall y \in K\}$. Note that $\ker(K)$ is a convex subset of $K$ and $\ker(K) = K$ if and only if $K$ is convex. For a parameter $\gamma > 0$, we define the following subset of $\mathcal{S}^d$ consisting of *well-conditioned* star bodies with nondegenerate kernels:

$$\mathcal{S}^d(\gamma) := \{K \in \mathcal{S}^d : \gamma B^d \subseteq \ker(K)\}.$$

Here $B^d := \{x \in \mathbb{R}^d : \|x\|_{\ell_2} \leqslant 1\}$ is the unit (Euclidean) ball in $\mathbb{R}^d$.

We finally discuss metric aspects of the space of star bodies. While the space of convex bodies enjoys the natural Hausdorff metric, there is a different, yet more natural metric over the space of star bodies dubbed the radial metric. Concretely, let $K, L \subset \mathbb{R}^d$ be star bodies. We define the *radial sum* $\tilde{+}$ between $K$ and $L$ as $K \tilde{+} L := \{x + y : x \in K, y \in L, x = \lambda y\}$; that is, unlike the Minkowski sum, we restrict the pair of vectors to be parallel. The radial sum obeys the relationship $\rho_{K \tilde{+} L}(u) := \rho_K(u) + \rho_L(u)$. We denote the *radial* metric between two star bodies $K, L$ as

$$\delta(K, L) := \inf\{\varepsilon \geqslant 0 : K \subseteq L \tilde{+} \varepsilon B^d, L \subseteq K \tilde{+} \varepsilon B^d\}.$$

In a similar fashion to how the support function induces the Haussdorf metric for convex bodies, the radial metric satisfies $\delta(K, L) := \|\rho_K - \rho_L\|_\infty$.

A functional $F : \mathcal{S}^d \to \mathbb{R}$ on the space of star bodies is continuous if it is continuous with respect to the radial metric, i.e., $F(\cdot)$ is continuous if for every $K \in \mathcal{S}^d$ and $(K_i)$ such that $\delta(K_i, K) \to 0$ as $i \to \infty$, then $F(K_i) \to F(K)$ as $i \to \infty$. Moreover, a subset $\mathfrak{C} \subset \mathcal{S}^d$ is closed if it is closed with respect to the topology induced by the radial metric. Finally, a subset $\mathfrak{C} \subset \mathcal{S}^d$ is bounded if there exists an $0 < R < \infty$ such that $K \subseteq RB^d$ for every $K \in \mathfrak{C}$.

## B.2   Proofs for Section 2

We require some preliminary results for our proofs. The first is a simple extension of a result in [52] to show continuity of our objective functional.

**Lemma B.1.** *Let $\mathcal{D}_r$ and $\mathcal{D}_n$ be distributions over $\mathbb{R}^d$ such that $\mathbb{E}_{\mathcal{D}_i}[\|x\|_{\ell_2}] < \infty$ for $i = r, n$. Then for any $K, L \in \mathcal{S}^d(\gamma)$, we have*

$$|F(K; \mathcal{D}_r, \mathcal{D}_n) - F(L; \mathcal{D}_r, \mathcal{D}_n)| \leqslant \frac{\mathbb{E}_{\mathcal{D}_r}[\|x\|_{\ell_2}] + \mathbb{E}_{\mathcal{D}_r}[\|x\|_{\ell_2}]}{\gamma^2} \delta(K, L).$$

*Proof of Lemma B.1.* We first note via the triangle inequality that

$$|F(K; \mathcal{D}_r, \mathcal{D}_n) - F(L; \mathcal{D}_r, \mathcal{D}_n)| \leqslant |\mathbb{E}_{\mathcal{D}_r}[\|x\|_K] - \mathbb{E}_{\mathcal{D}_r}[\|x\|_L]| + |\mathbb{E}_{\mathcal{D}_n}[\|x\|_K] - \mathbb{E}_{\mathcal{D}_n}[\|x\|_L]|.$$

Then the result follows from Corollary 2 in [52]. □

We additionally need the following local compactness result over the space of star bodies.

**Theorem B.2** (Theorem 5 in [52]). *Fix $0 < \gamma < \infty$ and let $\mathfrak{C}$ be a bounded and closed subset of $\mathcal{S}^d(\gamma)$. Let $(K_i)$ be a sequence of star bodies in $\mathfrak{C}$. Then $(K_i)$ has a subsequence that converges in the radial and Hausdorff metric to a star body $K \in \mathfrak{C}$.*

*Proof of Theorem 2.1.* The condition $\mathbb{E}_{\mathcal{D}_r}[\|x\|_{\ell_2}], \mathbb{E}_{\mathcal{D}_n}[\|x\|_{\ell_2}] < \infty$ guarantees that the expectation of $\|x\|_K$ exists for any $K \in \mathcal{S}^d(1)$. To see this, note that since $K \in \mathcal{S}^d(1)$, we have that $B^d \subseteq \ker(K)$ so that $\|x\|_K \leqslant \|x\|_{\ell_2}$ for all $x \in \mathbb{R}^d$. Thus

$$\mathbb{E}_{\mathcal{D}_i}[\|x\|_K] \leqslant \mathbb{E}_{\mathcal{D}_i}[\|x\|_{\ell_2}] < \infty \text{ for } i = r, n.$$

To prove the existence of minimizers, we first make the following useful connection regarding the minimization problem and the Wasserstein distance. Use the shorthand notation $F(K) := F(K; \mathcal{D}_r, \mathcal{D}_n)$. Recall the dual Kantorovich formulation of the 1-Wasserstein distance [92]:

$$W_1(\mathcal{D}_r, \mathcal{D}_n) = \sup_{f \in \mathrm{Lip}(1)} \mathbb{E}_{\mathcal{D}_n}[f(x)] - \mathbb{E}_{\mathcal{D}_r}[f(x)].$$

Since $\{\|\cdot\|_K : K \in \mathcal{S}^d(1)\} \subset \mathrm{Lip}(1)$, we see that

$$
\begin{aligned}
\inf_{K \in \mathcal{S}^d(1)} F(K) &= \inf_{\|\cdot\|_K \in \mathrm{Lip}(1)} \mathbb{E}_{\mathcal{D}_r}[\|x\|_K] - \mathbb{E}_{\mathcal{D}_n}[\|x\|_K] \\
&\geqslant \inf_{f \in \mathrm{Lip}(1)} \mathbb{E}_{\mathcal{D}_r}[f(x)] - \mathbb{E}_{\mathcal{D}_n}[f(x)] \\
&= \sup_{f \in \mathrm{Lip}(1)} \mathbb{E}_{\mathcal{D}_n}[f(x)] - \mathbb{E}_{\mathcal{D}_r}[f(x)] \\
&= W_1(\mathcal{D}_r, \mathcal{D}_n).
\end{aligned}
$$

This will prove to be useful in the existence proof.

To show the existence of a minimizer, we only consider the case when $W_1(\mathcal{D}_r, \mathcal{D}_n) > 0$ since if $W_1(\mathcal{D}_r, \mathcal{D}_n) = 0$, then any $K \in \mathcal{S}^d(1)$ is a minimizer of the objective since $F(K) = 0$ for all $K \in \mathcal{S}^d(1)$. Consider a minimizing sequence $(K_i) \subset \mathcal{S}^d(1)$ such that

$$F(K_i) \longrightarrow \inf_{K \in \mathcal{S}^d(1)} F(K) \text{ as } i \to \infty.$$

We claim that this sequence must be bounded, i.e., there exists an $R > 0$ such that $K_i \subseteq RB^d$ for all $i \geqslant 1$. Suppose this is not true. Then, for any $\ell = 1, 2, \ldots$, there exists a subsequence $(K_{i_\ell}) \subseteq (K_i)$ such that $K_{i_\ell} \supseteq \ell B^d$ for all $\ell$. That is, $K_{i_\ell}$ grows arbitrarily large as $\ell \to \infty$.

By Lemma B.1, we know that $F(\cdot)$ is continuous. Thus, we also must have that $F(K_{i_\ell}) \to \inf_{K \in \mathcal{S}^d(1)} F(K)$ as $\ell \to \infty$. But since $\ell B^d \subseteq K_{i_\ell}$, we must have that $\|x\|_{K_{i_\ell}} \leqslant \frac{1}{\ell}\|x\|_{\ell_2}$ for any $x \in \mathbb{R}^d$. This gives

$$\frac{1}{\ell} \mathbb{E}_{\mathcal{D}_r}[\|x\|_{\ell_2}] \geqslant \mathbb{E}_{\mathcal{D}_r}[\|x\|_{K_{i_\ell}}] \geqslant F(K_{i_\ell}) \geqslant \mathbb{E}_{\mathcal{D}_r}[\|x\|_{K_{i_\ell}}] - \frac{1}{\ell}\mathbb{E}_{\mathcal{D}_n}[\|x\|_{\ell_2}].$$

Taking the limit as $\ell \to \infty$ shows

$$\inf_{K \in \mathcal{S}^d(1)} F(K) = 0.$$

But this is a contradiction since

$$0 = \inf_{K \in \mathcal{S}^d(1)} F(K) \geqslant W_1(\mathcal{D}_r, \mathcal{D}_n) > 0$$

by assumption. Thus, we must have that the sequence $(K_i)$ is bounded, i.e., there exists an $R > 0$ such that $K_i \subseteq RB^d$ for all $i \geqslant 1$.

To complete the proof, observe that this minimizing sequence $(K_i) \subseteq \mathfrak{C} := \{K \in \mathcal{S}^d(1) : K_i \subseteq RB^d\}$, which is a closed and bounded subset of $\mathcal{S}^d(1)$. By Theorem B.2, this sequence must have a convergent subsequence $(K_{i_n})$ with limit $K_* \in \mathfrak{C} \subseteq \mathcal{S}^d(1)$. By continuity of $F(\cdot)$, this means

$$\lim_{i_n \to \infty} F(K_{i_n}) = F(K_*) = \inf_{K \in \mathcal{S}^d(1)} F(K)$$

as desired.

$\square$

*Proof of Theorem 2.4.* We first focus on proving the identity

$$\mathbb{E}_{\mathcal{D}_i}[\|x\|_K] = \int_{\mathbb{S}^{d-1}} \rho_{p_i}(u)^{d+1} \rho_K(u)^{-1} \mathrm{d}u \text{ for } i = r, n.$$

We prove this for $\mathcal{D}_r$, since the proof is identical for $\mathcal{D}_n$. Observe that for any $K \in \mathcal{S}^d$, since $\|u\|_K = \rho_K(u)^{-1}$, integrating in spherical coordinates gives

$$\begin{aligned}
\mathbb{E}_{\mathcal{D}_r}[\|x\|_K] = \int_{\mathbb{R}^d} \|x\|_K p_r(x) \mathrm{d}x &= \int_{\mathbb{S}^{d-1}} \int_0^\infty t^d \|u\|_K p_r(tu) \mathrm{d}t \mathrm{d}u \\
&= \int_{\mathbb{S}^{d-1}} \left( \int_0^\infty t^d p_r(tu) \mathrm{d}t \right) \|u\|_K \mathrm{d}u \\
&= \int_{\mathbb{S}^{d-1}} \rho_{p_r}(u)^{d+1} \rho_K(u)^{-1} \mathrm{d}u.
\end{aligned}$$

Now, applying this identity to our objective yields

$$\mathbb{E}_{\mathcal{D}_r}[\|x\|_K] - \mathbb{E}_{\mathcal{D}_n}[\|x\|_K] = \int_{\mathbb{S}^{d-1}} (\rho_{p_r}(u)^{d+1} - \rho_{p_n}(u)^{d+1}) \rho_K(u)^{-1} \mathrm{d}u.$$

Since $\rho_{p_r} > \rho_{p_n} \geqslant 0$ on the unit sphere, we have that the map $\rho_{r,n}(u) := (\rho_{p_r}(u)^{d+1} - \rho_{p_n}(u)^{d+1})^{1/(d+1)}$ is positive and continuous over the unit sphere. It is also positively homogenous

of degree $-1$, as both $\rho_{p_r}$ and $\rho_{p_n}$ are. Then, we claim that $\rho_{r,n}$ is the radial function of the unique star body

$$L_{r,n} := \{x \in \mathbb{R}^d : \rho_{r,n}(x)^{-1} \leqslant 1\}.$$

We first show that $L_{r,n}$ is star-shaped with gauge $\|x\|_{L_{r,n}} = \rho_{r,n}(x)^{-1}$. Note that the set is star-shaped since for each $x \in L_{r,n}$, we have for any $t \in [0,1]$,

$$\rho_{r,n}(tx)^{-1} = (\rho_{r,n}(x)/t)^{-1} = t\rho_{r,n}(x)^{-1} \leqslant t \leqslant 1$$

since $\rho_{r,n}$ is positively homogenous of degree $-1$. Hence $[0,x] \subseteq L_{r,n}$ for any $x \in L_{r,n}$. Then, we have that the gauge of $L_{r,n}$ satisfies

$$\begin{aligned}
\|x\|_{L_{r,n}} &= \inf\{t : x/t \in L_{r,n}\} = \inf\{t : \rho_{r,n}(x/t)^{-1} \leqslant 1\} \\
&= \inf\{t : \rho_{r,n}(x)^{-1} \leqslant t\} = \rho_{r,n}(x)^{-1}.
\end{aligned}$$

Hence the gauge of $L_{r,n}$ is precisely $\rho_{r,n}(x)^{-1}$. Moreover, by assumption, $u \mapsto \rho_{r,n}(u)$ is positive and continuous over the unit sphere so $L_{r,n}$ is a star body and is uniquely defined by $\rho_{r,n}$.

Combining the above results, we have the dual mixed volume interpretation

$$\mathbb{E}_{\mathcal{D}_r}[\|x\|_K] - \mathbb{E}_{\mathcal{D}_n}[\|x\|_K] = \int_{\mathbb{S}^{d-1}} \rho_{r,n}(u)^{d+1} \rho_K(u)^{-1} \mathrm{d}u = d\tilde{V}_{-1}(L_{r,n}, K).$$

Armed with this identity, we can apply Theorem 2.3 to obtain

$$\mathbb{E}_{\mathcal{D}_r}[\|x\|_K] - \mathbb{E}_{\mathcal{D}_n}[\|x\|_K] \geqslant d\operatorname{vol}_d(L_{r,n})^{(d+1)/d} \operatorname{vol}_d(K)^{-1/d}$$

with equality if and only if $L_{r,n}$ and $K$ are dilates. Hence the objective is minimized over the collection of unit-volume star bodies when $K_* := \operatorname{vol}_d(L_{r,n})^{-1/d} L_{r,n}$.

$\square$

## B.3 Proofs for Section 3

*Proof of Theorem 3.1.* Let $\rho_{\mathcal{D}_r}(u)$ denote the function induced by $\mathcal{D}_r$ via equation (3). We first understand $\mathbb{E}_{\mathcal{D}_n}[\|x\|_K^{-1}]$ :

$$\begin{aligned}
\mathbb{E}_{\mathcal{D}_n}[\|x\|_K^{-1}] &= \int_{\mathbb{R}^d} \|x\|_K^{-1} p_n(x) \mathrm{d}x \\
&= \int_{\mathbb{S}^{d-1}} \int_0^\infty t^{d-2} p_n(tu) \mathrm{d}t \|u\|_K^{-1} \mathrm{d}u \\
&= \int_{\mathbb{S}^{d-1}} \tilde{\rho}_{\mathcal{D}_n}(u)^{d-1} \|u\|_K^{-1} \mathrm{d}u \\
&= \int_{\mathbb{S}^{d-1}} \tilde{\rho}_{\mathcal{D}_n}(u)^{d-1} \rho_K(u) \mathrm{d}u
\end{aligned}$$

where here we have defined $\tilde{\rho}_{\mathcal{D}_n}(u) := \left(\int_0^\infty t^{d-2} p_n(tu) \mathrm{d}t\right)^{1/(d-1)}$ . Then, we can further write

$$\begin{aligned}
\mathbb{E}_{\mathcal{D}_r}[\|x\|_K] + \mathbb{E}_{\mathcal{D}_n}[\|x\|_K^{-1}] &= \int_{\mathbb{S}^{d-1}} \rho_K(u)^{-1} \rho_{\mathcal{D}_r}(u)^{d+1} \mathrm{d}u + \int_{\mathbb{S}^{d-1}} \rho_K(u) \tilde{\rho}_{\mathcal{D}_n}(u)^{d-1} \mathrm{d}u \\
&= \int_{\mathbb{S}^{d-1}} \rho_K(u)^{-1} \rho_{\mathcal{D}_r}(u)^{d+1} \mathrm{d}u + \int_{\mathbb{S}^{d-1}} \rho_K(u) \frac{\tilde{\rho}_{\mathcal{D}_n}(u)^{d-1}}{\rho_{\mathcal{D}_r}(u)^{d+1}} \rho_{\mathcal{D}_r}(u)^{d+1} \mathrm{d}u \\
&= \int_{\mathbb{S}^{d-1}} \left(\rho_K(u)^{-1} + \rho_K(u) \frac{\tilde{\rho}_{\mathcal{D}_n}(u)^{d-1}}{\rho_{\mathcal{D}_r}(u)^{d+1}}\right) \rho_{\mathcal{D}_r}(u)^{d+1} \mathrm{d}u \\
&=: d\tilde{V}_{-1}(L_r, K_{r,n})
\end{aligned}$$

where the star body $K_{r,n}$ is defined by its radial function

$$u \mapsto \left(\rho_K(u)^{-1} + \rho_K(u) \frac{\tilde{\rho}_{\mathcal{D}_n}(u)^{d-1}}{\rho_{\mathcal{D}_r}(u)^{d+1}}\right)^{-1}.$$

Note that $K_{r,n}$ is the $L_1$ *harmonic radial combination* of $K$ and the star body with radial function $u \mapsto \frac{\rho_{\mathcal{D}_r}(u)^{d+1}}{\rho_K(u)\tilde{\rho}_{\mathcal{D}_n}(u)^{d-1}}$. Let $L_r$ denote the star body with $\rho_{\mathcal{D}_r}$ as its radial function and $\tilde{L}_n$ denote the star body with radial function $\tilde{\rho}_{\mathcal{D}_n}$. The equality in the lower bound on $\tilde{V}_{-1}(K_{r,n}, L_r)$ is achieved by a star body $K$ such that for some $\lambda > 0$,

$$\rho_K(u)^{-1} + \rho_K(u)\frac{\rho_{\tilde{L}_n}(u)^{d-1}}{\rho_{L_r}(u)^{d+1}} = \frac{1}{\lambda\rho_{L_r}(u)}.$$

This is equivalent to

$$\lambda\frac{\rho_{\tilde{L}_n}(u)^{d-1}}{\rho_{L_r}(u)^d}\rho_K(u)^2 - \rho_K(u) + \lambda\rho_{L_r}(u) = 0.$$

For notational simplicity, for $u \in \mathbb{S}^{d-1}$, let $a(u) = \rho_{L_r}(u)$ and $b(u) = \rho_{\tilde{L}_n}(u)$. Then for $\lambda > 0$,

$$K_{r,n} = \lambda L_r \iff \lambda\frac{b(u)^{d-1}}{a(u)^d}\rho_K(u)^2 - \rho_K(u) + \lambda a(u) = 0, \ \forall u \in \mathbb{S}^{d-1}.$$

To solve for $\rho_K(u)$, we must analyze the roots of the polynomial $f(x) := \lambda\frac{b(u)^{d-1}}{a(u)^d}x^2 - x + \lambda a(u)$. But its roots are given by

$$x_* := x_*(u) := \frac{1 \pm \sqrt{1 - 4\lambda^2\left(\frac{b(u)}{a(u)}\right)^{d-1}}}{\frac{2\lambda b(u)^{d-1}}{a(u)^d}}.$$

Recall the requirement that $\rho_K$ must be positive and continuous over the unit sphere. For each $u \in \mathbb{S}^{d-1}$, $f$ has 2 positive roots if the discriminant is positive and 1 positive root if it is zero. Otherwise, both roots are complex. Hence, in order to have positive solutions for each $u$, we must have that

$$1 - 4\lambda^2\left(\frac{b(u)}{a(u)}\right)^{d-1} \geqslant 0, \forall u \in \mathbb{S}^{d-1} \iff 0 < \lambda \leqslant \sqrt{\frac{1}{4}\left(\frac{a(u)}{b(u)}\right)^{d-1}} \forall u \in \mathbb{S}^{d-1}.$$

Set

$$\lambda_* := \lambda_*(\mathcal{D}_r, \mathcal{D}_n) := \min_{u \in \mathbb{S}^{d-1}} \sqrt{\frac{1}{4}\left(\frac{a(u)}{b(u)}\right)^{d-1}} > 0.$$

Observe that $\lambda_*$ is positive and finite since $\rho_{L_r}$ and $\rho_{\tilde{L}_n}$ are positive and continuous over the unit sphere. Then for any $\lambda \in (0, \lambda_*]$, there are two possible star bodies $K_{+,\lambda}, K_{-,\lambda}$ that satisfy the above equation:

$$\rho_{K_{+,\lambda}}(u) := \frac{\rho_{L_r}(u)^d}{2\lambda\rho_{\tilde{L}_n}(u)^{d-1}}\left(1 + \sqrt{1 - 4\lambda^2\left(\frac{\rho_{\tilde{L}_n}(u)}{\rho_{L_r}(u)}\right)^{d-1}}\right)$$

$$\rho_{K_{-,\lambda}}(u) := \frac{\rho_{L_r}(u)^d}{2\lambda\rho_{\tilde{L}_n}(u)^{d-1}}\left(1 - \sqrt{1 - 4\lambda^2\left(\frac{\rho_{\tilde{L}_n}(u)}{\rho_{L_r}(u)}\right)^{d-1}}\right).$$

Notice that for any $u_*$ that achieves the minimum in the definition of $\lambda_*$, we have that $\rho_{K_{+,\lambda_*}}(u_*) = \rho_{K_{-,\lambda_*}}(u_*)$.

$\square$

### B.3.1 A general result for $\alpha$-divergences

As discussed in the main body of the paper, we are interested in deriving critic-based loss functions inspired by variational representations of divergences between probability measures. For general convex functions $f$, there is a well-known variational representation of the $f$-divergence $D_f$ defined via its convex conjugate $f^*(y) := \sup_{x \in \mathbb{R}} xy - f(x)$.

**Proposition 1** (Theorem 7.24 in [69]). *For $f$ convex, let $f^*$ denote its convex conjugate and $\Omega_*$ denote the effective domain of $f^*$. Then any $f$-divergence has the following variational representation*

$$D_f(P||Q) = \sup_{g:\mathbb{R}^d \to \Omega_*} \mathbb{E}_P[g] - \mathbb{E}_Q[f^* \circ g].$$

When applying this result to $\alpha$-divergences with $f(x) = f_\alpha(x) := \frac{x^\alpha - \alpha x - (1-\alpha)}{\alpha(\alpha-1)}$, we saw that it admits the representation (4). In particular, for $\alpha \in (-\infty, 0) \cup (0, 1)$, we can consider analyzing the following loss:

$$\alpha^{-1} \mathbb{E}_{\mathcal{D}_r}[\|x\|_K^\alpha] + (1-\alpha)^{-1} \mathbb{E}_{\mathcal{D}_n}[\|x\|_K^{\alpha-1}].$$

One can show that this loss can be written in terms of dual mixed volumes. When $\alpha \in (0, 1)$, the loss can be written as a single dual mixed volume. We will define the following notation, which will be useful throughout the proof: for a distribution $\mathcal{D}$ with density $p_\mathcal{D}$ and $\beta \in \mathbb{R}$, we set

$$\rho_{\beta,\mathcal{D}}(u) := \left( \int_0^\infty t^{d+\beta-1} p_\mathcal{D}(tu) \mathrm{d}t \right)^{\frac{1}{d+\beta}}.$$

We prove the following:

**Theorem 5.** *Fix $\alpha \in (-\infty, 0) \cup (0, 1)$. For two distributions $\mathcal{D}_r$ and $\mathcal{D}_n$ with densities $p_r$ and $p_n$, suppose that $\rho_{\alpha,\mathcal{D}_r}$ and $\rho_{\alpha-1,\mathcal{D}_n}$ and are positive and continuous over the unit sphere. Let $L_r^\alpha$ and $L_n^\alpha$ denote the star bodies with radial functions $\rho_{\alpha,\mathcal{D}_r}$ and $\rho_{\alpha-1,\mathcal{D}_n}$, respectively. Then there exists star bodies $\tilde{L}_r^\alpha$ and $\tilde{L}_n^\alpha$ that depend on $\mathcal{D}_r$, $\mathcal{D}_n$, and $K$ such that*

$$\alpha^{-1} \mathbb{E}_{\mathcal{D}_r}[\|x\|_K^\alpha] + (1-\alpha)^{-1} \mathbb{E}_{\mathcal{D}_n}[\|x\|_K^{\alpha-1}] = \alpha^{-1} d\tilde{V}_{-1}(\tilde{L}_r^\alpha, K) + (1-\alpha)^{-1} d\tilde{V}_{-1}(\tilde{L}_n^\alpha, K).$$

*If $\alpha \in (0, 1)$, then there exists a star body $K_{r,n}^\alpha$ that depends on $\mathcal{D}_r$, $\mathcal{D}_n$, and $K$ such that*

$$\alpha^{-1} \mathbb{E}_{\mathcal{D}_r}[\|x\|_K^\alpha] + (1-\alpha)^{-1} \mathbb{E}_{\mathcal{D}_n}[\|x\|_K^{\alpha-1}] = d\tilde{V}_{-\alpha}(K_{r,n}^\alpha, L_r^\alpha)$$

*where $\tilde{V}_{-\alpha}(L, K)$ is the $-\alpha$-dual mixed volume between $L$ and $K$. Moreover, we have the inequality*

$$\tilde{V}_{-\alpha}(L_r^\alpha, K_{r,n}^\alpha) \geqslant \mathrm{vol}_d(L_r^\alpha)^{(d+\alpha)/d} \mathrm{vol}_d(K_{r,n}^\alpha)^{-\alpha/d}$$

*with equality if and only if $K_{r,n}^\alpha$ is a dilate of $L_r^\alpha$. Any star body $K$ such that $K_{r,n}^\alpha$ is a dilate of $L_r^\alpha$ must satisfy the following: there exists a $\lambda > 0$ such that*

$$\alpha^{-1} \rho_K(u)^{-\alpha} + (1-\alpha)^{-1} \frac{\rho_{L_n^\alpha}(u)^{d+\alpha-1}}{\rho_{L_r^\alpha}(u)^{d+\alpha}} \rho_K(u)^{1-\alpha} = \lambda^{-1/\alpha} \rho_{L_r}(u)^{-1/\alpha} \text{ for all } u \in \mathbb{S}^{d-1}.$$

*Proof of Proposition 5.* Consider the loss

$$\alpha^{-1} \mathbb{E}_{\mathcal{D}_r}[\|x\|_K^\alpha] + (1-\alpha)^{-1} \mathbb{E}_{\mathcal{D}_n}[\|x\|_K^{\alpha-1}].$$

We first focus on $\mathbb{E}_{\mathcal{D}_r}[\|x\|_K^\alpha]$. Observe that

$$\begin{aligned}
\mathbb{E}_{\mathcal{D}_r}[\|x\|_K^\alpha] &= \int_{\mathbb{R}^d} \|x\|_K^\alpha p_r(x) \mathrm{d}x \\
&= \int_{\mathbb{S}^{d-1}} \int_0^\infty t^{d+\alpha-1} p_r(tu) \mathrm{d}t \|u\|_K^\alpha \mathrm{d}u \\
&= \int_{\mathbb{S}^{d-1}} \rho_{\alpha,\mathcal{D}_r}(u)^{d+\alpha} \|u\|_K^{\alpha-1} \|u\|_K \mathrm{d}u \\
&= \int_{\mathbb{S}^{d-1}} \rho_{\alpha,\mathcal{D}_r}(u)^{d+\alpha} \rho_K(u)^{1-\alpha} \rho_K(u)^{-1} \mathrm{d}u.
\end{aligned}$$

Let $\tilde{L}_r^\alpha$ be the star body with radial function

$$\rho_{\tilde{L}_r^\alpha}(u) := \left( \rho_{\alpha,\mathcal{D}_r}(u)^{d+\alpha} \rho_K(u)^{1-\alpha} \right)^{\frac{1}{d+1}}.$$

Indeed, this is a valid radial function since $\rho_{\tilde{L}_r^\alpha}$ is positively homogenous of degree $-1$ and is positive and continuous over the unit sphere. This shows that

$$\mathbb{E}_{\mathcal{D}_r}[\|x\|_K^\alpha] = \int_{\mathbb{S}^{d-1}} \rho_{\tilde{L}_r^\alpha}(u)^{d+1} \rho_K(u)^{-1} \mathrm{d}u = d\tilde{V}_{-1}(\tilde{L}_r^\alpha, K).$$

Similarly, we see that

$$
\begin{aligned}
\mathbb{E}_{\mathcal{D}_n}[\|x\|_K^{\alpha-1}] &= \int_{\mathbb{R}^d} \|x\|_K^{\alpha-1} p_n(x) \mathrm{d}x \\
&= \int_{\mathbb{S}^{d-1}} \int_0^\infty t^{d+\alpha-2} p_n(tu) \mathrm{d}t \|u\|_K^{\alpha-1} \mathrm{d}u \\
&= \int_{\mathbb{S}^{d-1}} \rho_{\alpha-1,\mathcal{D}_n}(u)^{d+\alpha-1} \|u\|_K^{\alpha-2} \|u\|_K \mathrm{d}u \\
&= \int_{\mathbb{S}^{d-1}} \rho_{\alpha-1,\mathcal{D}_n}(u)^{d+\alpha-1} \rho_K(u)^{2-\alpha} \rho_K(u)^{-1} \mathrm{d}u.
\end{aligned}
$$

Let $\tilde{L}_n^\alpha$ be the star body with radial function

$$
\rho_{\tilde{L}_n^\alpha}(u) := \left( \rho_{\alpha-1,\mathcal{D}_n}(u)^{d+\alpha-1} \rho_K(u)^{2-\alpha} \right)^{\frac{1}{d+1}}.
$$

This shows that

$$
\mathbb{E}_{\mathcal{D}_n}[\|x\|_K^{\alpha-1}] = \int_{\mathbb{S}^{d-1}} \rho_{\tilde{L}_n^\alpha}(u)^{d+1} \rho_K(u)^{-1} \mathrm{d}u = d\tilde{V}_{-1}(\tilde{L}_n^\alpha, K).
$$

For the final result, note that if $\alpha \in (0,1)$, then

$$
\begin{aligned}
\alpha^{-1} &\mathbb{E}_{\mathcal{D}_r}[\|x\|_K^\alpha] + (1-\alpha)^{-1} \mathbb{E}_{\mathcal{D}_n}[\|x\|_K^{\alpha-1}] \\
&= \int_{\mathbb{S}^{d-1}} \alpha^{-1} \rho_{\tilde{L}_r^\alpha}(u)^{d+1} \rho_K^{-1}(u) + (1-\alpha)^{-1} \rho_{\tilde{L}_n^\alpha}(u)^{d+1} \rho_K^{-1}(u) \mathrm{d}u \\
&= \int_{\mathbb{S}^{d-1}} \left( \alpha^{-1} \rho_{\alpha,\mathcal{D}_r}(u)^{d+\alpha} \rho_K(u)^{1-\alpha} + (1-\alpha)^{-1} \rho_{\alpha-1,\mathcal{D}_n}(u)^{d+\alpha-1} \rho_K(u)^{2-\alpha} \right) \rho_K(u)^{-1} \mathrm{d}u \\
&= \int_{\mathbb{S}^{d-1}} \rho_{\alpha,\mathcal{D}_r}(u)^{d+\alpha} \left( \alpha^{-1} \rho_K(u)^{-\alpha} + (1-\alpha)^{-1} \frac{\rho_{\alpha-1,\mathcal{D}_n}(u)^{d+\alpha-1}}{\rho_{\alpha,\mathcal{D}_r}(u)^{d+\alpha}} \rho_K(u)^{1-\alpha} \right) \mathrm{d}u \\
&=: d\tilde{V}_{-\alpha}(L_r^\alpha, K_{r,n}^\alpha)
\end{aligned}
$$

where we have defined the star body $K_{r,n}^\alpha$ via its radial function

$$
u \mapsto \left( \alpha^{-1} \rho_K(u)^{-\alpha} + (1-\alpha)^{-1} \frac{\rho_{\alpha-1,\mathcal{D}_n}(u)^{d+\alpha-1}}{\rho_{\alpha,\mathcal{D}_r}(u)^{d+\alpha}} \rho_K(u)^{1-\alpha} \right)^{-\alpha}
$$

and we also use the $-\alpha$-dual mixed volume, defined by

$$
\tilde{V}_{-\alpha}(L, K) := \frac{1}{d} \int_{\mathbb{S}^{d-1}} \rho_L(u)^{d+\alpha} \rho_K(u)^{-\alpha} \mathrm{d}u.
$$

The general dual mixed volume inequality of Lutwak (Theorem 2 in [55]) reads

$$
\tilde{V}_j^{k-i}(L, K) \leqslant \tilde{V}_i^{k-j}(L, K) \tilde{V}_k^{j-i}(L, K), \; i < j < k, i,j,k \in \mathbb{R}, \; L, K \in \mathcal{S}^d.
$$

Setting $i = -\alpha, j = 0, k = d$ gives

$$
\tilde{V}_{-\alpha}(L, K) \geqslant \mathrm{vol}_d(L)^{(d+\alpha)/d} \mathrm{vol}_d(K)^{-\alpha/d}
$$

with equality if and only if $K$ is a dilate of $L$. Applying this in our scenario gives

$$
\tilde{V}_{-\alpha}(L_r^\alpha, K_{r,n}^\alpha) \geqslant \mathrm{vol}_d(L_r^\alpha)^{(d+\alpha)/d} \mathrm{vol}_d(K_{r,n}^\alpha)^{-\alpha/d} \text{ with equality } \iff K_{r,n}^\alpha = \lambda L_r^\alpha.
$$

This only holds if there exists a $K$ and a $\lambda > 0$ such that the radial function of $K$ satisfies

$$
\left( \alpha^{-1} \rho_K(u)^{-\alpha} + (1-\alpha)^{-1} \frac{\rho_{L_n^\alpha}(u)^{d+\alpha-1}}{\rho_{L_r^\alpha}(u)^{d+\alpha}} \rho_K(u)^{1-\alpha} \right)^{-\alpha} = \lambda \rho_{L_r}(u) \text{ for all } u \in \mathbb{S}^{d-1}.
$$

$\square$

## B.4 Proofs for Section 4

*Proof of Proposition 4.2.* Suppose $M_{2,\rho}$ is convex. Then its gauge $\|x\|_{M_{2,\rho}}$ is a convex function. One can show that for any non-decreasing convex function $\phi$ and non-negative convex function $f$, the composition $\phi \circ f$ is convex. Setting $f(x) := \|x\|_{M_{2,\rho}}$ and $\phi(t) := t^2$ shows that $\|x\|_{M_{2,\rho}}^2$ is convex. Recalling that $\|x\|_{M_{2,\rho}}^2 = \|x\|_K^2 + \frac{\rho}{2}\|x\|_{\ell_2}^2$ implies that $\|x\|_K^2$ is $\rho$-weakly convex.

On the other hand, suppose $\|x\|_K^2$ is $\rho$-weakly convex. Then $\|x\|_{M_{2,\rho}}^2$ is convex. This implies that the level set $S_{\leqslant 1} := \{x \in \mathbb{R}^d : \|x\|_{M_{2,\rho}}^2 \leqslant 1\}$ is convex. But notice that $\|x\|_{M_{2,\rho}}^2 \leqslant 1$ if and only if $\|x\|_{M_{2,\rho}} \leqslant 1$ so

$$S_{\leqslant 1} = \{x \in \mathbb{R}^d : \|x\|_{M_{2,\rho}}^2 \leqslant 1\} = \{x \in \mathbb{R}^d : \|x\|_{M_{2,\rho}} \leqslant 1\} = M_{2,\rho}$$

which implies $M_{2,\rho}$ is convex.

$\square$

*Proof of Proposition 4.3.* Observe that $\mathcal{R}(\cdot)$ is positively homogenous as the composition of positively homogenous functions. Hence $x \mapsto 1/\mathcal{R}(x)$ is positively homogenous of degree $-1$. Then, we claim that

$$\min_{u \in \mathbb{S}^{d-1}} \mathcal{R}(u) > 0.$$

Note that since each $f_i$ is injective, positively homogenous, and continuous, it only vanishes at $0$ and is non-zero otherwise. Hence we have that for any $u \in \mathbb{S}^{d-1}$, $f_L \circ \cdots \circ f_1(u) \neq 0$ so that $\mathcal{R}(u) \neq 0$. Moreover, since $\mathcal{R}(\cdot)$ is continuous and $\mathbb{S}^{d-1}$ is compact, this minimum must be achieved by some $u_0 \in \mathbb{S}^{d-1}$. If $\mathcal{R}(u_0) = 0$, then this would contradict injectivity of each layer and positivity of $\mathcal{G}(\cdot)$. Hence $\mathcal{R}(u_0) > 0$. We conclude by noting $u \mapsto 1/\mathcal{R}(u)$ is continuous over $\mathbb{S}^{d-1}$ since $\mathcal{R}(u)$ is continuous as the composition of continuous functions and it is positive over $\mathbb{S}^{d-1}$. This guarantees $K$ is a star body.

Note that this proof also shows that $\mathcal{R}(\cdot)$ is coercive, since for any sequence $(x_k) \subset \mathbb{R}^d : \|x_k\|_{\ell_2} \to \infty$ as $k \to \infty$, we have by positive homogenity of $\mathcal{R}(\cdot)$,

$$\mathcal{R}(x_k) = \|x_k\|_{\ell_2} \mathcal{R}(x_k/\|x_k\|_{\ell_2}) \geqslant \|x_k\|_{\ell_2} \min_{u \in \mathbb{S}^{d-1}} \mathcal{R}(u) \to \infty \text{ as } k \to \infty.$$

$\square$

# C Experimental details

## C.1 Hellinger and adversarial loss comparison on MNIST

We provide additional implementation details for the experiments presented in Section 3.1. For our training data, we take 10000 random samples from the MNIST training set, constituting our $\mathcal{D}_r$ distribution. We then add Gaussian noise of variance $\sigma^2 = 0.05$ to each image, constituting our $\mathcal{D}_n$ distribution. The regularizers were parameterized via a deep convolutional neural network. Specifically, the network has 8 convolutional layers with LeakyReLU activation functions and an additional 3-layer MLP with LeakyReLU activations and no bias terms. The final layer is the Euclidean norm. This network implements a star-shaped regularizer, as outlined in the discussion of our Proposition 4.3. The regularizers were trained using the adversarial loss and Hellinger-based loss (5). We also used the gradient penalty term from [53] for both losses. We used the Adam optimizer for 20000 epochs and learning rate $10^{-3}$. The experiments were run on a single NVIDIA A100 GPU.

After training the regularizer $\mathcal{R}_\theta$, we then use it to denoise noisy *test* samples $y = x_0 + z$ where $x_0$ is a random test sample from the MNIST training set and $z \sim \mathcal{N}(0, \sigma^2 I)$ with $\sigma^2 = 0.05$. We do this by solving the following with gradient descent initialized at $y$:

$$\min_{x \in \mathbb{R}^d} \|x - y\|_{\ell_2}^2 + \lambda \mathcal{R}_\theta(x).$$

We ran gradient descent for 2000 iterations with a learning rate of $10^{-3}$. For the choice of regularization parameter $\lambda$, we note that in [53], the authors fix this value to be $\lambda := 2\tilde{\lambda}$ where

$\tilde{\lambda} := \mathbb{E}_{\mathcal{N}(0,\sigma^2 I)}\left[\|z\|_{\ell_2}\right]$ as the regularizer that achieves a small gradient penalty will be (approximately) 1-Lipschitz. For the Hellinger-based network, we found that $\lambda = 5.1\tilde{\lambda}^2$ gave better performance, so we used this for recovery. This may potentially be due to the fact that the regularizer learned via the Hellinger-based loss is less Lipschitz than the one learned via adversarial regularization. We additionally hypothesize this may also result from the fact that the Hellinger loss and adversarial loss weight the distributions $\mathcal{D}_r$ and $\mathcal{D}_n$ differently, resulting in different regularity properties of the learned regularizer. As such, we decided to additionally tune the regularization strength for the adversarially trained regularizer and found $\lambda = 0.75\tilde{\lambda}$ performed better than the original fixed value. Further studying these hyperparameter choices and how they are influenced by the loss function is an interesting direction for future study.

## C.2 Analysis on positive homogeneity

We conducted the same experiment as in Section 3.1. The main difference is that we changed the activation functions used in the network. Specifically, we use the same deep neural network as discussed in Section C.1, but considered the following four activation functions: LeakyReLU, Exponential Linear Unit (ELU), Tanh, and the Gaussian Error Linear Unit (GELU) activation. In Table 1, we show the average PSNR and MSE of each regularizer over the same 100 test images used in Section 3.1. We see that the Leaky ReLU-based regularizer outperforms regularizers using non-positively homogenous activation functions.

|            | PSNR  | MSE    |
|------------|-------|--------|
| Leaky ReLU | 22.32 | 0.0065 |
| ELU        | 20.74 | 0.0090 |
| Tanh       | 13.66 | 0.0431 |
| GELU       | 18.90 | 0.0157 |

Table 1: We show the average PSNR and MSE in recovering 100 test MNIST digits using regularizers trained via the adversarial loss. Each regularizer was parameterized by a convolutional neural network utilizing a single activation function from the following options: Leaky ReLU, ELU, Tanh, or GELU. The Leaky ReLU-based regularizer achieves the highest PSNR and lowest MSE.

# D  Additional results

## D.1 Stability results

We now illustrate stability results showcasing that if one learns star bodies over a finite amount of data, the sequence will exhibit a convergent subsequence whose limit will be a solution to the idealized population risk. We prove this result when the constraint set corresponds to star bodies of unit-volume with $\gamma B^d \subseteq \ker(K)$. Define

$$\mathcal{S}^d(\gamma, 1) := \{K \in \mathcal{S}^d : \gamma B^d \subseteq \ker(K),\ \mathrm{vol}_d(K) = 1\}.$$

A result due to [52] shows that this subset of star bodies is uniformly bounded in the sense that there exists an $R > 0$ such that every $K \in \mathcal{S}^d(\gamma, 1)$ satisfies $K \subseteq RB^d$.

**Lemma D.1** (Lemma 1 in [52]). *For any $\gamma > 0$, the collection $\mathcal{S}^d(\gamma, 1)$ is a bounded subset of $\mathcal{S}^d(\gamma)$. In particular, for $R_\gamma := \frac{d+1}{\gamma^{d-1}\kappa_{d-1}}$ where $\kappa_{d-1} := \mathrm{vol}_{d-1}(B^{d-1})$,*

$$\mathcal{S}^d(\gamma, 1) \subseteq \{K \in \mathcal{S}^d(\gamma) : K \subseteq R_\gamma B^d\}.$$

Now, to state the convergence result, for a distribution $\mathcal{D}$, let $\mathcal{D}^N$ denote the empirical distribution over $N$ i.i.d. draws from $\mathcal{D}$. The proof of this result uses tools from $\Gamma$-convergence [14] and local compactness properties of $\mathcal{S}^d(\gamma)$.

**Theorem D.2.** *Let $\mathcal{D}_r$ and $\mathcal{D}_n$ be distributions over $\mathbb{R}^d$ such that $\mathbb{E}_{\mathcal{D}_r}\|x\|_{\ell_2}, \mathbb{E}_{\mathcal{D}_n}\|x\|_{\ell_2} < \infty$ and fix $0 < \gamma < \infty$. Then the sequence of minimizers $(K_N^*) \subseteq \mathcal{S}^d(\gamma, 1)$ of $F(K; \mathcal{D}_r^N, \mathcal{D}_n^N)$ over $\mathcal{S}^d(\gamma, 1)$ has the property that any convergent subsequence converges in the radial and Hausdorff metric to a minimizer of the population risk almost surely:*

$$\text{any convergent } (K_{N_\ell}^*) \text{ satisfies } K_{N_\ell}^* \to K_* \in \underset{K \in \mathcal{S}^d(\gamma, 1)}{\arg\min}\ F(K; \mathcal{D}_r, \mathcal{D}_n).$$

*Moreover, a convergent subsequence of $(K_N^*)$ exists.*

To prove this result, we first state the definition of $\Gamma$-convergence here and cite some useful results that will be needed in our proofs.

**Definition D.3.** Let $(F_i)$ be a sequence of functions $F_i : X \to \mathbb{R}$ on some topological space $X$. Then we say that $F_i$ $\Gamma$-converges to a limit $F$ and write $F_i \xrightarrow{\Gamma} F$ if the following conditions hold:

- For any $x \in X$ and any sequence $(x_i)$ such that $x_i \to x$, we have

$$F(x) \leqslant \liminf_{i \to \infty} F_i(x_i).$$

- For any $x \in X$, we can find a sequence $x_i \to x$ such that

$$F(x) \geqslant \limsup_{i \to \infty} F_i(x_i).$$

In fact, if the first condition holds, then the second condition could be taken to be the following: for any $x \in X$, there exists a sequence $x_i \to x$ such that $\lim_{i \to \infty} F_i(x_i) = F(x)$.

In addition to $\Gamma$-convergence, we also require the notion of equi-coercivity, which states that minimizers of a sequence of functions are attained over a compact domain.

**Definition D.4.** A family of functions $F_i : X \to \mathbb{R}$ is equi-coercive if for all $\alpha$, there exists a compact set $K_\alpha \subseteq X$ such that $\{x \in X : F_i(x) \leqslant \alpha\} \subseteq K_\alpha$.

Finally, this notion combined with $\Gamma$-convergence guarantees convergence of minimizers, which is known as the Fundamental Theorem of $\Gamma$-convergence [14].

**Proposition D.5** (Fundamental Theorem of $\Gamma$-Convergence). *If $F_i \xrightarrow{\Gamma} F$ and the family $(F_i)$ is equi-coercive, then the every limit point of the sequence of minimizers $(x_i)$ where $x_i \in \arg\min_{x \in X} F_i(x)$ converges to some $x \in \arg\min_{x \in X} F(x)$.*

In order to show that our empirical risk functional $\Gamma$-converges to the population risk, we need to show that the empirical risk uniformly converges to the population risk as the amount of samples $N \to \infty$. We exploit the following result, which shows that uniform convergence is possible when the hypothesis class of functions admits $\varepsilon$-coverings.

**Theorem D.6** (Theorem 3 in [68]). *Let $Q$ be a probability measure, and let $Q_m$ be the corresponding empirical measure. Let $\mathcal{G}$ be a collection of $Q$-integrable functions. Suppose that for every $\varepsilon > 0$ there exists a finite collection of functions $\mathcal{G}_\varepsilon$ such that for every $g \in \mathcal{G}$ there exists $\overline{g}, \underline{g} \in \mathcal{G}_\varepsilon$ satisfying (i) $\underline{g} \leqslant g \leqslant \overline{g}$, and (ii) $\mathbb{E}_Q[\overline{g} - \underline{g}] < \varepsilon$. Then $\sup_{g \in \mathcal{G}} |\mathbb{E}_{Q_m}[g] - \mathbb{E}_Q[g]| \to 0$ almost surely.*

We now state and prove the uniform convergence result.

**Theorem D.7.** *Fix $0 < \gamma < \infty$ and let $\mathcal{D}_r$ and $\mathcal{D}_n$ be distributions on $\mathbb{R}^d$ such that $\mathbb{E}_{\mathcal{D}_i}\|x\|_{\ell_2} < \infty$ for $i = r, n$. Then, we have strong consistency in the sense that*

$$\sup_{K \in \mathcal{S}^d(\gamma, 1)} |F(K; \mathcal{D}_r^N, \mathcal{D}_n^N) - F(K; \mathcal{D}_r, \mathcal{D}_n)| \to 0 \text{ as } N \to \infty \text{ almost surely.}$$

*Proof of Theorem D.7.* By Lemma B.1, we have that the map $K \mapsto F(K; \mathcal{D}_r, \mathcal{D}_n)$ is $CM_P$-Lipschitz over $\mathcal{S}^d(\gamma, 1)$ where $C = C(\gamma) := 1/\gamma^2$ and $M_P := \max_{i=r,n} \mathbb{E}_{\mathcal{D}_i}\|x\|_{\ell_2} < \infty$. Now, consider the set of functions $\mathcal{G} := \{\|\cdot\|_K : K \in \mathcal{S}^d(\gamma, 1)\}$. We will show that we can construct a finite set of functions that approximate $\|\cdot\|_K$ for any $K \in \mathcal{S}^d(\gamma, 1)$ via an $\varepsilon$-covering argument. This will allow us to apply Theorem D.6. First, note that $\mathcal{S}^d(\gamma, 1)$ is closed (with respect to both

the radial and Hausdorff topology) and bounded as a subset of $\mathcal{S}^d(\gamma)$. Thus, by Theorem B.2, we have that the space $(\mathcal{S}^d(\gamma,1),\delta)$ is sequentially compact, as every sequence has a convergent subsequence with limit in $\mathcal{S}^d(\gamma,1)$. On metric spaces, sequential compactness is equivalent to compactness. Since the space is compact, it is totally bounded, thus guaranteeing the existence of a finite $\varepsilon$-net for every $\varepsilon > 0$ as desired. For fixed $\varepsilon > 0$, construct a $\eta$-cover $\mathcal{S}_\eta$ of $\mathcal{S}^d(\gamma,1)$ such that $\sup_{K\in\mathcal{S}^d(\gamma,1)} \min_{L\in\mathcal{S}_\eta} \delta(K,L) \leqslant \eta$ where $\eta \leqslant \varepsilon/(2CM_P)$. Define the following sets of functions:

$$\mathcal{G}_{\eta,-} := \{(\|\cdot\|_K - CM_P\eta)_+ : K \in \mathcal{S}_\eta\} \text{ and } \mathcal{G}_{\eta,+} := \{\|\cdot\|_K + CM_P\eta : K \in \mathcal{S}_\eta\}.$$

Let $\|\cdot\|_K \in \mathcal{G}$ be arbitrary. Let $K_0 \in \mathcal{S}_\eta$ be such that $\delta(K,K_0) \leqslant \eta$. Define $\overline{f} = \|\cdot\|_{K_0} + CM_P\eta \in \mathcal{G}_{\eta,+}$ and $\underline{f} = (\|\cdot\|_{K_0} - CM_P\eta)_+ \in \mathcal{G}_{\eta,-}$. It follows that $\underline{f} \leqslant f \leqslant \overline{f}$. Moreover, by our choice of $\eta$, we have that

$$\mathbb{E}_{\mathcal{D}_i}[\overline{f} - \underline{f}] \leqslant 2CM_P\eta \leqslant \varepsilon \text{ for each } i = r, n.$$

Thus, the conditions of Theorem D.6 are met and we get

$$\sup_{K\in\mathcal{S}^d(\gamma,1)} |F(K;\mathcal{D}_r^N,\mathcal{D}_n^N) - F(K;\mathcal{D}_r,\mathcal{D}_n)| \leqslant \sup_{K\in\mathcal{S}^d(\gamma,1)} |\mathbb{E}_{\mathcal{D}_r^N}[\|x\|_K] - \mathbb{E}_{\mathcal{D}_r}[\|x\|_K]|$$

$$+ \sup_{K\in\mathcal{S}^d(\gamma,1)} |\mathbb{E}_{\mathcal{D}_n^N}[\|x\|_K] - \mathbb{E}_{\mathcal{D}_n}[\|x\|_K]|$$

$$\to 0 \text{ as } N \to \infty \text{ a.s.}$$

$\square$

## D.2 Proof of Theorem D.2

We first establish the two requirements of $\Gamma$-convergence. For the first, fix $K \in \mathcal{S}^d(\gamma,1)$ and consider a sequence $K_N \to K$. Then we have that

$$F(K;\mathcal{D}_r,\mathcal{D}_n) = F(K;\mathcal{D}_r,\mathcal{D}_n) - F(K;\mathcal{D}_r^N,\mathcal{D}_n^N) + F(K;\mathcal{D}_r^N,\mathcal{D}_n^N)$$

$$- F(K_N;\mathcal{D}_r^N,\mathcal{D}_n^N) + F(K_N;\mathcal{D}_r^N,\mathcal{D}_n^N)$$

$$\leqslant |F(K;\mathcal{D}_r,\mathcal{D}_n) - F(K;\mathcal{D}_r^N,\mathcal{D}_n^N)|$$

$$+ |F(K;\mathcal{D}_r^N,\mathcal{D}_n^N) - F(K_N;\mathcal{D}_r^N,\mathcal{D}_n^N)| + F(K_N;\mathcal{D}_r^N,\mathcal{D}_n^N)$$

$$\leqslant \sup_{K\in\mathcal{S}^d(\gamma,1)} |F(K;\mathcal{D}_r,\mathcal{D}_n) - F(K;\mathcal{D}_r^N,\mathcal{D}_n^N)|$$

$$+ |F(K;\mathcal{D}_r^N,\mathcal{D}_n^N) - F(K_N;\mathcal{D}_r^N,\mathcal{D}_n^N)| + F(K_N;\mathcal{D}_r^N,\mathcal{D}_n^N). \tag{6}$$

By Theorem D.7, we have that the first term goes to 0 as $N$ goes to $\infty$ almost surely. We now show that $|F(K;\mathcal{D}_r^N,\mathcal{D}_n^N) - F(K_N;\mathcal{D}_r^N,\mathcal{D}_n^N)| \to 0$ as $N \to \infty$ almost surely. To see this, observe that by Lemma B.1 we have

$$|F(K;\mathcal{D}_r^N,\mathcal{D}_n^N) - F(K_N;\mathcal{D}_r^N,\mathcal{D}_n^N)| \leqslant \frac{\max_{i=r,n}\mathbb{E}_{\mathcal{D}_i^N}[\|x\|_{\ell_2}]}{r^2}\delta(K,K_N).$$

Since $\mathbb{E}_{\mathcal{D}_i^N}[\|x\|_{\ell_2}] \to \mathbb{E}_{\mathcal{D}_i}[\|x\|_{\ell_2}] < \infty$ for each $i = r, n$ and $\delta(K,K_N) \to 0$ as $N \to \infty$ almost surely, we attain $|F(K;\mathcal{D}_r^N,\mathcal{D}_n^N) - F(K_N;\mathcal{D}_r^N,\mathcal{D}_n^N)| \to 0$. Thus, taking the limit inferior of both sides in equation (6) yields

$$F(K;\mathcal{D}_r,\mathcal{D}_n) \leqslant \liminf_{m\to\infty} F(K_N;\mathcal{D}_r^N,\mathcal{D}_n^N).$$

For the second requirement, we exhibit a realizing sequence so let $K \in \mathcal{S}^d(\gamma,1)$ be arbitrary. By the proof of Theorem D.7, for any $N \geqslant 1$, there exists a finite $\frac{1}{N}$-net $\mathcal{S}_{1/N}$ of $\mathcal{S}^d(\gamma,1)$ in the radial metric $\delta$. Construct a sequence $(K_N) \subset \mathcal{S}^d(\gamma,1)$ such that for each $N$, $K_N \in \mathcal{S}_{1/N}$ and satisfies $\delta(K_N,K) \leqslant 1/N$. Hence this sequence satisfies $K_N \to K$ in the radial metric and $K_N \in \mathcal{S}^d(\gamma,1)$ for all $N \geqslant 1$. Hence we can apply Theorem D.7 to get

$$|F(K_N;\mathcal{D}_r^N,\mathcal{D}_n^N) - F(K;P)| \leqslant |F(K_N;\mathcal{D}_r^N,\mathcal{D}_n^N) - F(K;\mathcal{D}_r^N,\mathcal{D}_n^N)|$$

$$+ |F(K;\mathcal{D}_r^N,\mathcal{D}_n^N) - F(K;P)|$$

$$\to 0 \text{ as } N \to \infty \text{ a.s.}$$

so $\lim_{N\to\infty} F(K_N; \mathcal{D}_r^N, \mathcal{D}_n^N) = F(K; \mathcal{D}_r, \mathcal{D}_n)$.

Now, we show that $(F(\cdot; \mathcal{D}_r^N, \mathcal{D}_n^N))$ is equi-coercive on $\mathcal{S}^d(\gamma, 1)$. In fact, this follows directly from Theorem B.2, the variant of Blaschke's Selection Theorem we proved for the radial metric. Thus equi-coerciveness of the family $(F(\cdot; \mathcal{D}_r^N, \mathcal{D}_n^N))$ trivially holds over $\mathcal{S}^d(\gamma, 1)$. As a result, applying Proposition D.5 to the family $F(\cdot; \mathcal{D}_r^N, \mathcal{D}_n^N) : \mathcal{S}^d(\gamma, 1) \to \mathbb{R}$, if we define the sequence of minimizers

$$K_N^* \in \arg\min_{K \in \mathcal{S}^d(\gamma,1)} F(K; \mathcal{D}_r^N, \mathcal{D}_n^N)$$

we have that any limit point of $K_N^*$ converges to some

$$K_* \in \arg\min_{K \in \mathcal{S}^d(\gamma,1)} F(K; \mathcal{D}_r, \mathcal{D}_n)$$

almost surely, as desired. The existence of a convergent subsequence of $(K_N^*)$ follows from $\mathcal{S}^d(\gamma, 1)$ being a closed and bounded subset of $\mathcal{S}^d(\gamma)$ and Theorem B.2.

### D.3  Further examples for Section 2

**Example 6** ($\ell_1$- and $\ell_2$-ball example)**.** *We additionally plot the underlying sets $L_r$ and $L_n^\alpha$ from the example in the main body in Figure 5.*

**Example 7** ($\ell_\infty$-ball and a Gaussian)**.** *We consider the following case for $d = 2$, when our distributions $\mathcal{D}_r$ and $\mathcal{D}_n$ are given by a Gibbs density of the $\ell_\infty$-norm and a Gaussian with mean $0$ and covariance $\Sigma \in \mathbb{R}^{d\times d}$ :*

$$p_r(x) = \frac{1}{\Gamma(d+1)\,\mathrm{vol}_d(2B_{\ell_\infty})}e^{-\frac{1}{2}\|x\|_{\ell_\infty}} \text{ and } p_n^\alpha(x) = \frac{1}{\sqrt{(2\pi)^d \det(\Sigma)}}e^{-\frac{1}{2}\langle x, \Sigma^{-1}x\rangle}.$$

*Here, we will set $\Sigma = [0.1, 0.3; 0, 0.1] \in \mathbb{R}^{2\times 2}$. The star bodies induced by $p_r$ and $p_n^\alpha$ are dilations of the $\ell_1$-ball and the ellipsoid induced by $\Sigma$, respectively. Denote these star bodies by $L_r$ and $L_n$, respectively. Then, the data-dependent star body $L_{r,n}$ is defined by*

$$\rho_{L_{r,n}^\alpha}(u) := \left(c_r\|x\|_{\ell_1}^{-(d+1)} - c_n\|\Sigma^{-1/2}x\|_{\ell_2}^{-(d+1)}\right)^{1/(d+1)}$$

*where the constants $c_r := \int_0^\infty t^d \exp(-t)/(\Gamma(d+1)\,\mathrm{vol}_d(2B_{\ell_\infty}))\mathrm{d}t$ and $c_n := \frac{1}{\sqrt{(2\pi)^d \det(\Sigma)}}\int_0^\infty t^d \exp(-\frac{1}{2}t^2)\mathrm{d}t$. We visualize the star bodies in Figure 6.*

**Example 8** (Densities induced by star bodies)**.** *We give general examples of $L_{r,n}$ when the distributions $\mathcal{D}_r$ and $\mathcal{D}_n$ have densities induced by star bodies. If we assume the conditions of Theorem 2.4, using the definition of $\rho_{p_r}$ and $\rho_{p_n}$, we have that radial function $\rho_{r,n}$ is of the form*

$$\rho_{r,n}(u)^{d+1} = \int_0^\infty t^d(p_r(tu) - p_n(tu))\mathrm{d}t.$$

*Suppose for some functions $\psi_r, \psi_n$ such that $\int_0^\infty t^d\psi_i(t)\mathrm{d}t < \infty$ for $i = r, n$, we have $p_r(u) = \psi_r(\|x\|_{K_r})$ and $p_n(u) = \psi_n(\|x\|_{K_n})$ for two star bodies $K_r$ and $K_n$. Then we have that*

$$\int_0^\infty t^d(p_r(tu) - p_n(tu))\mathrm{d}t = c(\psi_r)\|x\|_{K_r}^{-(d+1)} - c(\psi_n)\|x\|_{K_n}^{-(d+1)}$$

*where $c(\psi) := \int_0^\infty t^d\psi(t)\mathrm{d}t$. Note then we must have*

$$c(\psi_r)\|x\|_{K_r}^{-(d+1)} > c(\psi_n)\|x\|_{K_n}^{-(d+1)} \Leftrightarrow c(\psi_r)^{-1/(d+1)}\|x\|_{K_r} < c(\psi_n)^{-1/(d+1)}\|x\|_{K_n}.$$

*This effectively requires the containment property*

$$c(\psi_n)^{1/(d+1)}K_n \subset c(\psi_r)^{1/(d+1)}K_r.$$

**Example 9** (Exponential densities)**.** *Suppose, for simplicity, that we have exponential densities. We need them to integrate to 1, so, for example, letting $c_d := \Gamma(d+1)$, we know from [52] that $\int_{\mathbb{R}^d} e^{-\|x\|_K}\mathrm{d}x = c_d\,\mathrm{vol}_d(K)$. Hence suppose our two distributions have densities $p_r(x) = e^{-\|x\|_{K_r}}/(c_d\,\mathrm{vol}_d(K_r))$ and $p_n(x) = e^{-\|x\|_{K_n}}/(c_d\,\mathrm{vol}_d(K_n))$. Then $\int_{\mathbb{R}^d} p_r(x)\mathrm{d}x =$*

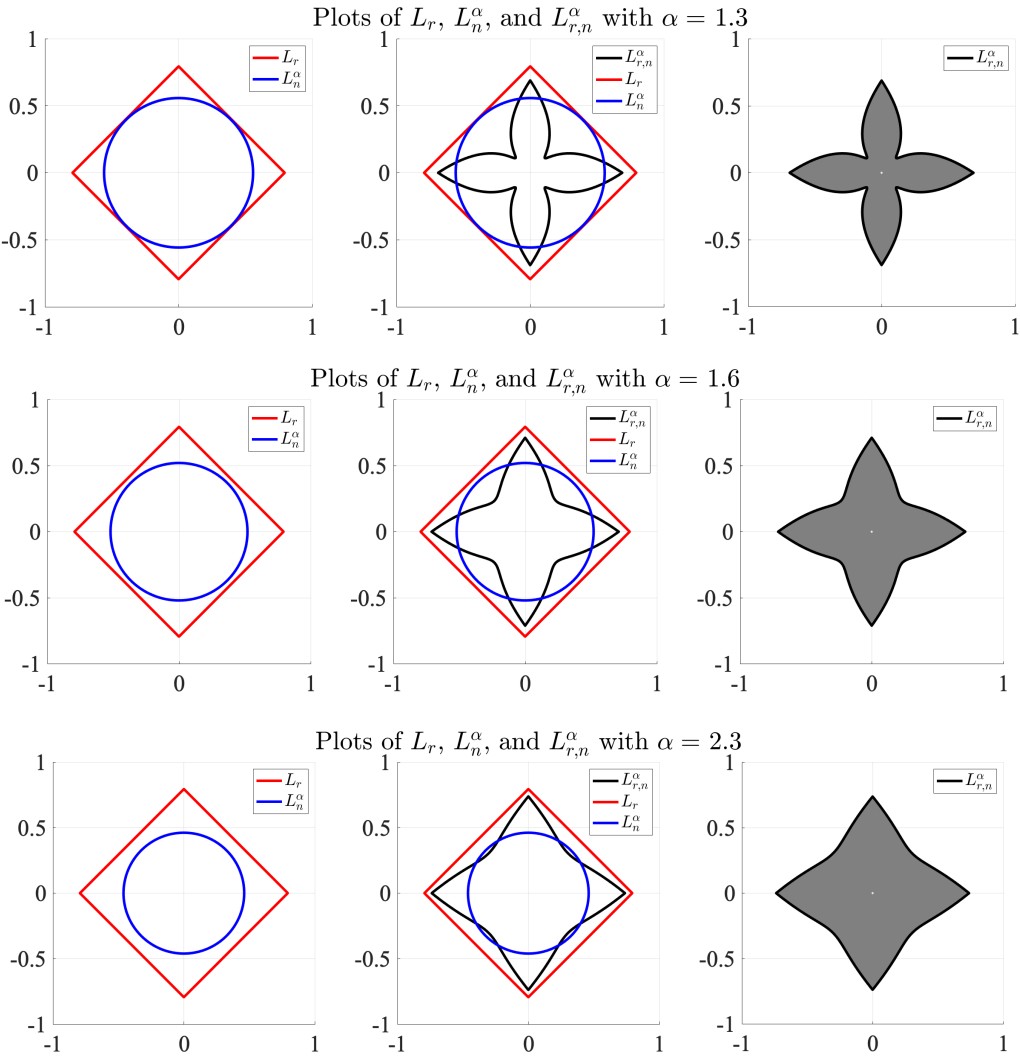

Figure 5: We plot the sets $L_r$, $L_n^\alpha$, and $L_{r,n}^\alpha$ for different values of $\alpha$: (Top) $\alpha = 1.3$, (Middle) $\alpha = 1.6$, and (Bottom) $\alpha = 2.3$. In each row, the left figure shows the boundaries of $L_r$ and $L_n^\alpha$, the middle figure additionally overlays the boundary of $L_{r,n}^\alpha$ and the right figure shows $L_{r,n}^\alpha := \{x \in \mathbb{R}^2 : \|x\|_{L_{r,n}^\alpha} \leqslant 1\}$.

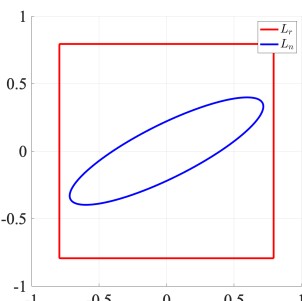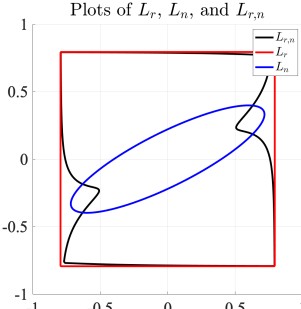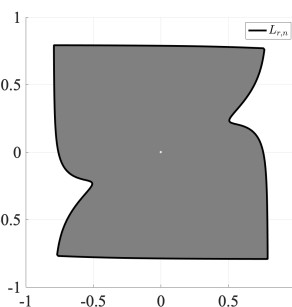

Plots of $L_r$, $L_n$, and $L_{r,n}$

Figure 6: We consider the example where $L_r$ is induced by a Gibbs density with the $\ell_\infty$-norm and $L_n$ is induced by a mean 0 Gaussian distribution with covariance $\Sigma = [0.1, 0.3; 0, 0.1]$. (Left) We show the boundaries of $L_r$ and $L_n$. (Middle) We additionally overlay the boundary of $L_{r,n}$. (Right) We show $L_{r,n} := \{x \in \mathbb{R}^2 : \|x\|_{L_{r,n}} \leqslant 1\}$.

$\int_{\mathbb{R}^d} p_n(x)\mathrm{d}x = 1$. *Here,* $\psi_r(t) = e^{-t}/(c_d \operatorname{vol}_d(K_r))$ *and* $\psi_n(t) = e^{-t}/(c_d \operatorname{vol}_d(K_n))$. *Hence, in order for the distributions* $\mathcal{D}_r$ *and* $\mathcal{D}_n$ *to satisfy the assumptions of Theorem 2.4, we require*

$$\frac{\rho_{K_r}(u)^{d+1}}{\operatorname{vol}_d(K_r)} > \frac{\rho_{K_n}(u)^{d+1}}{\operatorname{vol}_d(K_n)} \text{ for all } u \in \mathbb{S}^{d-1}.$$

*In the end, our star body will have radial function of the form*

$$\rho_{r,n}(u)^{d+1} = \tilde{c}_d \left( \frac{\rho_{K_r}(u)^{d+1}}{\operatorname{vol}_d(K_r)} - \frac{\rho_{K_n}(u)^{d+1}}{\operatorname{vol}_d(K_n)} \right)$$

*for some constant* $\tilde{c}_d$ *that depends on the dimension.*

**Example 10** (Scaling star bodies). *Using the previous example, we can see how scaling a star body allows one to satisfy the conditions of the theorem. In particular, suppose we have exponential densities where* $K_r = \alpha K_n$ *where* $\alpha > 1$. *Then note that the above requirement for containment will be satisfied since for any* $u \in \mathbb{S}^{d-1}$,

$$\rho_{K_r}(u)^{d+1}/\operatorname{vol}_d(K_r) = \alpha^{d+1}\rho_{K_n}(u)^{d+1}/(\alpha^d \operatorname{vol}_d(K_n))$$
$$= \alpha \rho_{K_n}(u)^{d+1}/\operatorname{vol}_d(K_n)$$
$$> \rho_{K_n}(u)^{d+1}/\operatorname{vol}_d(K_n).$$

The above example shows that for certain distributions, we are always able to scale them in such a way that they will always satisfy the assumptions of our Theorem. Below we give a more general result along these lines.

**Proposition D.8.** *Suppose* $K_r$ *and* $K_n$ *are two star bodies in* $\mathbb{R}^d$. *Then, there exists constants* $0 < m < M < \infty$ *(depending on* $K_r$ *and* $K_n$*) such that if we set*

$$\alpha_* := \frac{1}{2} \cdot \frac{\operatorname{vol}_d(K_n)}{\operatorname{vol}_d(K_r)} \left( \frac{m}{M} \right)^{d+1} > 0$$

*we have that*

$$\frac{\rho_{K_r}(u)^{d+1}}{\operatorname{vol}_d(K_r)} > \frac{\rho_{\alpha_* K_n}(u)^{d+1}}{\operatorname{vol}_d(\alpha_* K_n)}, \ \forall u \in \mathbb{S}^{d-1}.$$

*That is, for any two star bodies* $K_r, K_n \in \mathcal{S}^d$, *the assumptions of Theorem 2 will be satisfied with the distributions* $\mathcal{D}_r$ *and* $\mathcal{D}_n$ *with densities*

$$p_r(x) := \frac{e^{-\|x\|_{K_r}}}{c_d \operatorname{vol}_d(K_r)} \text{ and } p_n(x) := \frac{e^{-\|x\|_{\alpha_* K_n}}}{c_d \operatorname{vol}_d(\alpha_* K_n)},$$

*respectively.*

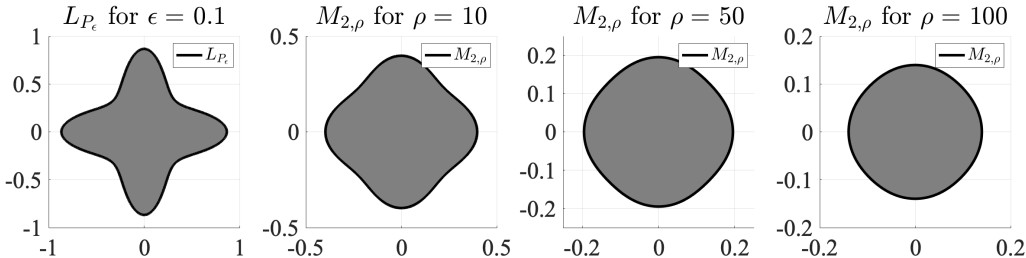

Figure 7: (First) The Gaussian mixture model in Section D.4 induces a star body $L_{P_\varepsilon}$ Here, we set $\varepsilon = 0.1$. Smaller values of $\varepsilon$ induce a higher degree of nonconvexity in $L_{P_\varepsilon}$. In the next three images, we show the resulting harmonic 2-combination $M_{2,\rho} := L_{P_\varepsilon} \hat{+}_2 \frac{\rho}{2} B^d$ for (Second) $\rho = 10$, (Third) $\rho = 50$, and (Fourth) $\rho = 100$. We see that for some value of $\rho_*(\varepsilon) := \rho_* > 10$, $M_{2,\rho_*}$ becomes convex. By Proposition 4.2, $x \mapsto \|x\|^2_{L_{P_\varepsilon}}$ will be $\rho_*$-weakly convex.

*Proof of Proposition D.8.* Because $K_n$ and $K_r$ are star bodies, they are bounded with the origin in their interior. Hence, there exists constants $0 < m < M < \infty$ such that $mB^d \subseteq K_n, K_r \subseteq MB^d$. Note that in order for us to have

$$\rho_{K_r}(u)^{d+1}/\operatorname{vol}_d(K_r) > \rho_{\alpha K_n}(u)^{d+1}/\operatorname{vol}_d(\alpha K_n) = \alpha\rho_{K_n}(u)^{d+1}/\operatorname{vol}_d(K_n) \ \forall\, u \in \mathbb{S}^{d-1}$$

we require

$$\alpha < \frac{\operatorname{vol}_d(K_n)}{\operatorname{vol}_d(K_r)} \left(\frac{\rho_{K_r}(u)}{\rho_{K_n}(u)}\right)^{d+1} \ \forall\, u \in \mathbb{S}^{d-1}.$$

But by our assumptions on $K_r$ and $K_n$, note that $\rho_{K_r} \geqslant \rho_{mB^d} = m\rho_{B_d} = m$ and also $\rho_{K_n} \leqslant \rho_{MB^d} = M\rho_{B^d}$ on the sphere. This follows by the monotonicity property of the radial function for star bodies: $K \subseteq L \iff \rho_K \leqslant \rho_L$. Thus, using these bounds achieves the desired result.

$\square$

### D.4 A star body with weakly convex squared gauge

We will consider an example of a data-dependent, nonconvex star body such that its squared gauge is weakly convex for a sufficiently large parameter $\rho > 0$. For a parameter $\varepsilon \in (0,1)$, consider the following 2-dimensional Gaussian mixture model $P_\varepsilon = \frac{1}{2}\mathcal{N}(0, \Sigma_{\varepsilon,1}) + \frac{1}{2}\mathcal{N}(0, \Sigma_{\varepsilon,2})$ where $\Sigma_{\varepsilon,1} := [1,0;0,\varepsilon] \in \mathbb{R}^{2\times 2}$ and $\Sigma_{\varepsilon,2} := [\varepsilon,0;0,1] \in \mathbb{R}^{2\times 2}$. Denote its density by $p_\varepsilon(x)$. One can show that $p_\varepsilon$ induces a valid radial function via equation (3) given by

$$\rho_{P_\varepsilon}(u) = \left(\int_0^\infty r^2 p_\varepsilon(ru)\mathrm{d}u\right)^{1/3} = \left(c_{\varepsilon,1}\|\Sigma_{\varepsilon,1}^{-1/2}u\|_{\ell_2}^{-3} + c_{\varepsilon,2}\|\Sigma_{\varepsilon,2}^{-1/2}u\|_{\ell_2}^{-3}\right)^{1/3}$$

where $c_{\varepsilon,i} = \frac{1}{2}\det(2\pi\Sigma_{\varepsilon,i})^{-1/2}\left(\int_0^\infty t^2 e^{-t^2/2}\mathrm{d}t\right)$ for $i = 1, 2$. Let $L_{P_\varepsilon}$ denote the star body with radial function $\rho_{P_\varepsilon}$. For a fixed $\varepsilon \in (0,1)$, we investigate the geometry of $M_{2,\rho} := L_{P_\varepsilon} \hat{+}_2 \frac{\rho}{2} B^d$ for various values of $\rho$ in Figure 7. In this example, we set $\varepsilon = 0.1$. As discussed in [52], $\check{L}_{P_\varepsilon}$ can be described as the harmonic Blaschke linear combination between the star bodies induced by the distributions $\mathcal{N}(0, \Sigma_{\varepsilon,1})$ and $\mathcal{N}(0, \Sigma_{\varepsilon,2})$. These distributions induce ellipsoids that are concentrated on one of the axes, and their resulting harmonic Blaschke linear combination is the nonconvex star body shown in the left panel in Figure 7. We see that as $\rho$ increases, the resulting harmonic 2-combination $M_{2,\rho}$ eventually becomes convex for a large enough parameter $\rho_*(\varepsilon) := \rho_*$ that depends on $\varepsilon$. By Proposition 4.2, this shows that the squared gauge $\|\cdot\|^2_{L_{P_\varepsilon}}$ is $\rho_*$-weakly convex.

