# OpenReview forum: "The Star Geometry of Critic-Based Regularizer Learning"
_NeurIPS.cc/2024/Conference — NeurIPS 2024 poster_

### Official Review · Reviewer_kSE2 · 2024-07-11

**Soundness:** 3
**Presentation:** 4
**Contribution:** 4
**Rating:** 8
**Confidence:** 3

**Summary:**

The paper presents a theoretical analysis of learning regularizers for inverse problems using a critic-based loss. By focusing on a specific family of regularizers (gauges of star-shaped bodies), amenable to theoretical analysis, the authors provide a number of theoretical insights towards existence, uniqueness within existing frameworks (based on wasserstein distance) and further extensions to f-divergences. This is further connected to the existing literature on learned regularization by considering star bodies corresponding to weakly convex regularisers.

**Strengths:**

The paper presents a very novel idea of utilising the theoretical framework of star-bodies in order to provide theoretical interpretability of critic based regularization. Given the recent interest in learned regularization, this paper opens up a number of new research directions both theoretically and numerically.

**Weaknesses:**

There are a few weaknesses, which in my opinion are not limiting. To be precise, the paper focuses on a specific class of regularizers (gauges of star-shaped bodies) and a specific type of critic-based loss functions (derived from variational representations of statistical distances). It would be interesting to see if the results can be extended to other classes of regularizers and loss functions.

The paper primarily investigates this class of regularizers theoretically and with very few experiments. The paper does not include any experiments to demonstrate the practical performance of the learned regularizers, and while the theoretical results are valuable, it would be helpful to see how they translate into practice. This could be of interest as future work for practitioners working on inverse problems.

**Questions:**

The paper is very well-writen, and as such there are very few questions that I have:

* One of the motivations, also discussed in 1.2 (and line 57), is that uniqueness of the transport potential does not hold when considering the wasserstein 1 based loss. I would like to refer the authors to arxiv.org/abs/2211.00820, as in fact (under some assumptions), this uniqueness can be shown to be unique $D_n$ almost everywhere. With this result in mind, could you explain intuitively why in Theorem 2.4, it is possible to prove uniqueness without a.e.?

* Line 130 "this map" - which map is this referring to?

* I am not entirely sure what the relevance of Remark 2.8 is. In practice, rescaling the distribution destroys information from the true distribution - the critic that is desired is the one that would be operating on $D_r$ and $D_n$, and not $D_r$ and $\lambda D_n$.

* It would be very intersting to see whether the optimal regularisers derived as minimisers of the variational objective are also optimal regulariser in the sense of Leong et al.

*Line 50 "about the measurements". Clasically the measurements themselves live in a different space from the original data. For this reason Lunz et al. utilises backprojection to first map it to the same space.

* Line 29 "ill-posed meaning that there are an infinite number ..." - In the inverse problem literature, ill-posedness does not correspond to non-uniqueness only. I suggest referring to Hadamards definition of well-posedness. E.g. see Shumaylov et al. or Arridge, Simon, et al. "Solving inverse problems using data-driven models." Acta Numerica 28 (2019): 1-174.

**Limitations:**

* See weaknesses.

---

> ### Author Rebuttal · Authors · 2024-08-06
>
> We thank the reviewer for their detailed review and positive comments that our paper is "very well-written" and that our theoretical framework "opens up a number of new research directions both theoretically and numerically." Below we discuss some of the main questions that were raised.
>
> >How do we obtain uniqueness
>
> Thank you for raising this point. The reason why Theorem 2.4 does not require an "almost everywhere" stipulation is that our optimization problem is over the class of star body gauges, as opposed to star-shaped gauges, or a more general function class. At a high-level, star body gauges have additional structure that allow for us to obtain uniqueness, which can be lost when one considers more general star-shaped gauges.
>
> In more detail, star bodies are in one-to-one correspondence with their radial functions (or the reciprocal of their gauge) and are uniquely defined by them. As a result, dual mixed volumes also exactly specify star bodies, e.g., for star bodies $K,\tilde{K}, L,$ and $\tilde{L}$, we have $\tilde{V}(K,\tilde{K}) = \tilde{V}(L,\tilde{L})$ if and only if $K = L$ and $\tilde{K} = \tilde{L}$ (see Lutwak [52]). This property aids in establishing uniqueness in our main Theorem.
>
> If we relax the requirement of being a star body to simply being star-shaped, we can lose such a property. In particular, if we consider gauges of star-shaped sets, it could be possible for the dual mixed volumes of different sets to be equal because two star-shaped gauges can be equal "almost everywhere". For example, consider the following two star-shaped sets in $\mathbb{R}^2$: $K = B_{\ell_2}$ is the unit $\ell_2$ ball, and $L = B_{\ell_2} \cup \set{(0,t) : t \geqslant 0}$. Both sets are star-shaped with respect to the origin, but $K$ is a star body while $L$ is not. This is because the ray $\set{t e_2 : t \geqslant 0}$ where $e_2 = (0,1)$ intersects the boundary of $L$ infinitely many times. The gauges of $K$ and $L$ are equal to $||x||_{\ell_2}$ for any $x$ outside of the set of measure zero $\set{(0,t) : t \geqslant 0}.$ This is because for any $(0,t)$ with $t > 0$, $||(0,t)||_K = |t|$ while $||(0,t)||_L = 0$.
>
> >Line 130
>
> Our apologies, "this map" refers to the map $x \mapsto ||x||_K$. We will fix this in the manuscript.
>
> >Remark 2.8
>
> We thank the reviewer for noting this. Our goal in this remark was to show that if the distributions do not satisfy the assumptions of the theorem, there is a way to reweight the objective so that the assumptions of the theorem are satisfied. By positive homogeneity of the gauge, this essentially amounts to reweighting/scaling one of the terms $\lambda\mathbb{E}_{\mathcal{D}_i}[||x||_K]$ for some $\lambda \geqslant 0$. We will update the discussion in the manuscript to make this more clear.
>
> >Line 50
>
> Thank you for pointing this out. We intended to use the phrase "integrate information about the measurements" in a vague sense, meaning that the "bad" distribution $\mathcal{D}_n$ depends on $y$ in some way. We will update this phrasing to avoid this confusion.
>
> >Line 29
>
> Thank you for making this point. Indeed, ill-posedness additionally encompasses a lack of a solution's existence and whether it varies continuously in the data. We will update this in the revised manuscript and cite Arridge et al's survey.

---

> > ### Comment · Reviewer_kSE2 · 2024-08-09
> >
> > Thanks to the authors for their clarifications and answers. I would suggest adding a note about the 'almost everywhere' as a distinguishing factor from the rest of the literature. I would be very happy to see the paper accepted.

---

> > > ### Author Response · Authors · 2024-08-09
> > >
> > > Thank you to the reviewer for taking the time to consider our rebuttal and for the suggestion. We agree that this is a good point to highlight, and will be sure to do so in a revised version of the manuscript. We sincerely appreciate the reviewer's support of our paper.

---

### Official Review · Reviewer_XeHQ · 2024-07-13

**Soundness:** 3
**Presentation:** 2
**Contribution:** 3
**Rating:** 6
**Confidence:** 2

**Summary:**

This paper explores the learning of task-dependent regularizers using critic-based loss functions in the context of variational regularization for statistical inference and inverse problems. It particularly focuses on a specific family of regularizers, namely gauges of star-shaped bodies, which are common in practice and similar to those parameterized by deep neural networks. The study introduces a novel approach utilizing tools from star geometry and dual Brunn-Minkowski theory, which allows the derivation of exact expressions for the optimal regularizer under certain conditions and explores the properties of neural network architectures that can yield such regularizers. This work contributes to a deeper understanding of the structure of data-driven regularizers and their optimization characteristics.

**Strengths:**

The problem setup and the theoretical framework of the paper appear rigorous and methodologically sound. The motivation behind the study is robust, addressing the theoretical gaps in understanding how regularizers are learned.

**Weaknesses:**

Some of the results presented in the paper are complex and difficult to interpret, which may limit their accessibility to a broader audience. Moreover, the paper does not clearly articulate the practical implications of these theoretical findings for real-world applications, which could hinder its impact.

**Questions:**

1. Could you provide additional context on the adversarial regularization framework, particularly regarding the roles and definitions of the two distributions $D_r$ and $D_n$?

2. The significance of the results in Theorem 3.1 is not clear to me. Could you elaborate on why these results are important and what they contribute to the field of regularizer learning?

3. The paper lacks experimental validation of its theoretical constructs through simulations or empirical data. Some experiments (even simple ones) will be helpful for better understanding.

**Limitations:**

Yes.

---

> ### Author Rebuttal · Authors · 2024-08-06
>
> We thank the reviewer for their detailed review and positive comments that our theoretical framework is "rigorous and methodologically sound" and that our results "address the theoretical gaps in understanding how regularizers are learned." Below we discuss some of the main questions that were raised.
>
> >Additional context on the adversarial regularization framework
>
> In the adversarial regularization framework of Lunz et al [51], the distributions $\mathcal{D}_r$ and $\mathcal{D}_n$ are user-specified distributions that are aimed to capture “real” or “good” data and “noisy” or “bad” data, respectively. The choice of these distributions depends on the specific application one is interested in. In the context of inverse problems, $\mathcal{D}_r$ is often chosen to be the distribution of images similar to the image we wish to reconstruct and $\mathcal{D}_n$ is the distribution of noisy or poor reconstructions. For example, suppose one is interested in solving an inverse problem $y = \mathcal{A}(x) + \eta$ where $\mathcal{A}$ is known and $\eta$ is drawn from some noise distribution $\mathcal{P}$. Suppose we have prior knowledge about $x$ that it is drawn from a distribution $\mathcal{D}_r$ and we have access to many samples from this distribution (e.g., we want to solve inpainting problems on human faces, and we have access to many samples from the CelebA dataset). One choice for $\mathcal{D}_n$ corresponds to the distribution of backprojected reconstructions, which corresponds to taking one's measurements and "naively" inverting them using the pseudoinverse of $\mathcal{A}$: $\hat{x} \sim \mathcal{D}_n$ if and only if $\hat{x} = \mathcal{A}^{\dagger} y$ where $y = \mathcal{A}(x) + \eta$, $x \sim \mathcal{D}_r$, and $\eta \sim \mathcal{P}$. These images will be corrupted by noise that depends on both the measurement matrix $\mathcal{A}$ but also the additive noise $\eta$.
>
> This backprojected distribution is a good choice for the "bad" distribution for two reasons. One is that the distribution contains noise that is pertinent to the inverse problem (i.e., the samples depend on $\eta$ and $\mathcal{A}$). Another reason is that backprojected reconstructions are often used as the initial iterate for gradient-based methods to solve
>
> $$\min_{x \in \mathbb{R}^d} \frac{1}{2}|| y - \mathcal{A}(x)||_2^2 + \mathcal{R}(x).$$
> Hence, if the regularizer $\mathcal{R}(x)$ has been learned to assign very low likelihood to backprojected reconstructions, using them as initial iterates to a gradient-based algorithm would aid in the algorithm making good progress initially, as the regularizer would give gradients that would push the algorithm away from these poor, noisy solutions.
>
> We will add this additional discussion as background on the adversarial regularization framework in an updated version of the manuscript.
>
> >The significance of Theorem 3.1
>
> Thank you for raising this point. At a high level, Theorem 3.1 is significant and has implications for regularizer learning for three reasons:
> - This theorem introduces and analyzes a novel critic-based loss function for learning regularizers. To our knowledge, the only other critic-based loss function that has been considered in the literature is the loss from the adversarial regularization framework of Lunz et al [51]. Introducing novel loss criteria to learn regularizers is broadly useful for the field of regularizer learning, and opens new avenues for future research, both from a theoretical and practical perspective.
> - The theorem offers novel insights into the structure of regularizers one would learn from this loss. We give explicit expressions for their structure and are able to compare and contrast such regularizers with those found via the adversarial regularization framework of Lunz et al [51]. This can be seen by comparing the visual examples shown in Figures 2 and 4.
> - Analyzing this loss brings new challenges and technical contributions for both the field of star geometry and regularizer learning. Please see our global comment for more details on the new technical challenges that this theorem addresses.

---

> > ### Author Response · Authors · 2024-08-12
> >
> > Dear reviewer,
> >
> > Thank you again for your detailed review and positive comments. We wanted to provide you with a brief update on our work. Based on your and other reviewer's comments, we have conducted new experiments to support our theory. Please see the official comment titled "New Experimental Results". We hope that these experiments along with our rebuttal can potentially address your concerns.

---

> > ### Comment · Reviewer_XeHQ · 2024-08-13
> >
> > Thank you for answering my questions. I keep my original score.

---

> > > ### Author Response · Authors · 2024-08-13
> > >
> > > We thank the reviewer for considering our rebuttal and keeping their positive score.

---

### Official Review · Reviewer_3HXp · 2024-07-13

**Soundness:** 3
**Presentation:** 3
**Contribution:** 1
**Rating:** 5
**Confidence:** 4

**Summary:**

This paper leverages the star geometry and dual Brunn-Minkowski theory to study the optimal critic-based regularizers. The authors illustrate the optimal regularizer can be interpreted using dual mixed volumes that depend on data distribution. Theorems are proved for the existence and uniqueness of the optimal regularizer. The authors also identify the neural network architectures for learning the star body gauges for the optimal regularizer.

**Strengths:**

The paper leverages the star geometry in understanding the geometry of unsupervised regularizer learning.

**Weaknesses:**

As cited in the submission, this paper is closely related to [50]. Many concepts, theoretic results, even examples resemble or coincide with those in [50]. The paper failed to clearly distinguish itself from the existing work [50].

**Questions:**

In what scope the submitted work extends [50]?
On line 68, "assigns" should be "assigned"?
On line 118, what does "[x, y]" mean if both x, y are points in R^d? Did you mean the line segment connecting x and y?

**Limitations:**

See the weakness

---

> ### Author Rebuttal · Authors · 2024-08-06
>
> We thank the reviewer for their comments and questions. Below we would like to address concerns/questions that were raised.
>
> >Regarding novelty in relation to [50]
>
> We appreciate the reviewer's concern regarding novelty in relation to [50]. Please see our global comment regarding the novelty and significance of our work. The following points summarize our work's impact and novelty:
> - Our research introduces novel theory and a new framework for understanding critic-based regularizer learning, addressing a significant gap in existing theoretical foundations.
> - We tackle several novel technical challenges inherent to the critic-based setting, distinguishing our work from [50].
> - This work also demonstrates the applicability and utility of star geometry tools for regularizer learning and broader machine learning contexts.
>
> We would also like to highlight that while our results and example look similar to those in [50], all results in the present work are novel. Moreover, all examples we visualize in the paper are different than those in [50]. The only example that is similar is the one shown in Figure 7 in the supplemental. Our experiment is different, however, since we are showing that this star body induces a weakly convex (squared) gauge, which was not explored in [50].
>
> >Typos and notation
>
> Thank you for your catching those typos. We will update them in our revised submission. Yes, $[x,y]$ refers to the line segment between these two points in $\mathbb{R}^d$, i.e., $[x,y] := \set{(1-t)x + ty : t \in [0,1]}.$ We will add this definition in the manuscript.

---

> > ### Comment · Reviewer_3HXp · 2024-08-08
> >
> > Thanks to the authors for their clarification. I see the contribution better and will raise my score.

---

> > > ### Author Response · Authors · 2024-08-09
> > >
> > > Thank you to the reviewer for taking the time to consider our rebuttal. We sincerely appreciate raising your score.

---

### Official Review · Reviewer_5WpE · 2024-07-15

**Soundness:** 3
**Presentation:** 3
**Contribution:** 3
**Rating:** 6
**Confidence:** 3

**Summary:**

This submission extends the techniques of [50], i.e., tools from star geometry and dual Brunn-Minkowski theory, to characterize the optimal regularizer under the adversarial regularization framework [51] of inverse problems. \alpha-Divergence as loss functions for learning regularizers is also discussed, with the dual mixed volume interpretations. The weak convexity and compatible neural network architectures are further discussed for computational concerns related to the proposed star body regularizers.

**Strengths:**

Extending the analysis and results of [50] to the adversarial regularization framework and showing its connections to the \alpha-divergence is an interesting theoretical contribution. The specified neural network layers compatible with the star body regularizer can also shed light on practice.

**Weaknesses:**

My major concern with this work is that it is unclear if the proposed new \alpha-divergence-based loss functions are useful for the adversarial regularization problem this submission studies. The original adversarial regularization work [51] for inverse problems, albeit published in NeurIPS 6 years ago, has reported experimental results to validate the proposed framework. On the other hand, if positioned as a pure theory work, given the existence of [50], I feel that the theoretical contribution of this submission seems a bit short for publication in NeurIPS.

As a minor thing, it would be helpful for readability if an overview of the organization and the flow of the paper could be briefly presented in the introduction.

After the authors' adding new experiments during the discussion phase
---------------------------------------------------------------------------------------------
These two concerns are partially addressed. On one hand, I can see the potential of the proposed approach from these experiments. On the other hand, the experiments are still preliminary and small scale.

**Questions:**

Prop. 4.3 requires positive homogeneity of each layer, which limits the choice of activation functions in practice to a subset of piecewise linear functions, such as ReLU and its variants. I feel that this could be a limitation of the proposed approach. Moreover, while each layer is an injective function, due to the homogeneity of activations, the composite of such layers will not be an injective (see Sec. 3 of the paper below for example), will this break the proof of Prop. 4.3?

Dinh, Laurent, et al. "Sharp minima can generalize for deep nets." ICML 2017.

After the authors' rebuttal
---------------------------------------------------------------------------------------------------
The question regarding the composite of injectivities does not apply to Prop. 4.3.

**Limitations:**

Mostly. Please refer to the Questions session for a concern I have regarding limitations.

---

> ### Author Rebuttal · Authors · 2024-08-06
>
> We thank the reviewer for their detailed review and positive comments that our insights offer "an interesting theoretical contribution". Below we would like to address concerns/questions that were raised.
>
> >Regarding novelty in relation to [50]
>
> We appreciate the reviewer's concern regarding novelty in relation to [50]. Please see our global comment above regarding the novelty and significance of our work. We would additionally like to note that the new proposed $\alpha$-divergence-based loss functions are meant to offer an alternative to the adversarial regularization framework that still falls under the umbrella of "critic-based" regularizer learning frameworks. While we did not provide experiments in this paper, we believe our theoretical contributions provide novel insights into unsupervised, critic-based regularizer learning, an area that has been underdeveloped theoretically, and offer convincing evidence that other critic-based losses are worth exploring both from a theoretical and practical perspective.
>
> >Adding an overview
>
> Thank you for this suggestion. We agree that this would help with readability of the manuscript. We will add this in an updated version of the manuscript.
>
> >Regarding Proposition 4.3
>
> Yes, we agree that requiring positive homogeneity of each layer limits the activation functions one can use. However, such activations are commonly employed in such tasks. For example, the experiments that were done in Lunz et al [51] and in subsequent applications also used a network of this form, i.e., a convolutional neural network with no biases and LeakyReLU activations.
>
> Regarding the concern of a lack of injectivity, we first note that Prop 4.3 requires that each layer $f_i(\cdot)$ is injective and positively homogenous. Note that the composition of any two injective and positively homogenous functions is again injective and positively homogenous. To see this, consider two injective and positively homogenous functions $f$ and $g$. Then note that for any $\lambda \geqslant 0$ and $x$, $f(g(\lambda x)) = f(\lambda g(x)) = \lambda f(g(x))$. To see injectivity, note that $g(x) = g(y)$ implies $x = y$. The same holds true for $f$. Hence if $f(g(x)) = f(g(y))$, then $g(x) = g(y)$ by injectivity of $f$. But by injectivity of $g$, $x = y$. Hence $f(g(\cdot))$ is injective.
>
> When applying this to neural networks, note that our result requires each layer $f_i(\cdot)$ to be injective and positively homogenous. This means that we require that the linear layer $W_i$ *composed with the activation function* $\sigma_i$ must be injective and positively homogenous, i.e., $f_i(\cdot) = \sigma_i(W\cdot)$ must be injective and positively homogenous. This is automatically satisfied if the linear layer is injective and the activation is LeakyReLU, since it is bijective. This can fail, however, if one uses ReLU. For example, note that $\mathrm{ReLU}(I \cdot)$ is not injective, even though the identity matrix $I$ is invertible. Certain matrices $W$, however, allow for injective maps $x \mapsto \mathrm{ReLU}(Wx)$. See, for example, the notion of a "directed spanning set" in Puthawala et al or the injective 1x1 convolutions of Kothari et al.
>
> Puthawala et al, Globally Injective ReLU Networks, Journal of Machine Learning Research 23 (2022) 1-55
>
> Kothari et al, TRUMPETS: Injective Flows for Inference and Inverse Problems, Uncertainty in Artificial Intelligence (2021)

---

> > ### Comment · Reviewer_5WpE · 2024-08-09
> >
> > Thanks to the authors for the detailed response to concerns and questions, which helps me to better access the novelty and theoretical contribution of this submission. As a result, I have raised my rating.
> >
> > My previous question regarding the composite of injectivities actually apply to the mappings from parameters to network functions, rather than the input-output network functions, so this does not apply to Prop. 4.3 and the authors' explanation is correct.
> >
> > With all that said, my concerns regarding limited activation function choices and lack of experimental validations remain. That is why I am not able to raise the rating to a firm accept for NeurIPS.

---

> > > ### Author Response · Authors · 2024-08-12
> > >
> > > Thank you to the reviewer for considering our rebuttal and for raising their rating. Based on your concerns, we conducted experiments on two points that were raised, namely the applicability of the $\alpha$-divergence based loss and the importance of the homogeneity of the activation function. Please see the official comment posted titled "New Experimental Results".
> > >
> > > Thank you for voicing your concerns and suggesting we explore this further. We believe that these additional experiments are valuable in providing support for our theory and will significantly improve the paper quality. Please let us know if you have any additional questions or comments. We hope that these experiments can potentially address your concerns.

---

> > > > ### Comment · Reviewer_5WpE · 2024-08-12
> > > >
> > > > I would like to thank the authors for conducting experiments regarding my two major concerns. While the experiments are quite preliminary and small-scale, I can see the potential of the proposed approach from them. Therefore, I will further raise my rating.

---

> > > > > ### Author Response · Authors · 2024-08-13
> > > > >
> > > > > Thank you to the reviewer for appreciating our experiments and acknowledging the practical potential of our theory. We sincerely appreciate raising your score.

---

### Author Rebuttal · Authors · 2024-08-06

We thank all reviewers for their detailed feedback and positive comments that our work is "rigorous and methodologically sound" and that our theory "opens up a number of new research directions both theoretically and numerically". We will address each reviewer's specific questions and concerns individually, but we would also like to address a main concern that was shared by multiple reviewers here. In particular, we would like to discuss the novelty and significance of our results in relation to [50]. We intend to update the manuscript with additional discussion highlighting the differences between our work and [50].

## Novelty in relation to [50]

We appreciate the reviewers' concerns regarding the novelty of our work. While it is true that our current paper employs similar mathematical tools to [50], we would like to highlight the following key differences and novel contributions:

**1. Novel setting and new theory for unsupervised regularizer learning**

The current work focuses on understanding critic-based regularization, an inherently different problem than the one considered in [50]. Critic-based regularizers must learn to prefer a pre-defined "clean/good" distribution over a "noisy/bad" distribution, whereas the setting in [50] asks to find a regularizer that maximizes the likelihood of the data. Understanding how these two distributions differ from one another creates novel challenges that make this setting inherently different than [50], a point that we will discuss further in our novel technical contributions.

Moreover, the theoretical foundations of critic-based regularization remain underdeveloped. To date, aside from the seminal work of Lunz et al. [51], there has been minimal theoretical exploration of the structure of regularizers learned in the critic-based setting. Our results demonstrate that star-geometric tools can help establish new theoretical frameworks for critic-based regularizer learning, addressing this notable gap in the literature.

**2. Novel technical contributions**

While our tools are superficially similar, the novel setting of critic-based regularizer learning brings new challenges that we believe are of interest to the regularizer learning community. Additionally, this work provides new analyses of star body gauges that were not present in [50]. We highlight specifics below:

- **New challenges**: As an example, the new critic-based loss for regularizer learning in equation (5) brings novel challenges in star geometry. In particular, we show that this loss has a dual mixed volume interpretation, but its dependency on $K$ is more complicated than in [50]. As a result, its extremizers no longer directly follow from the dual mixed volume inequality (as they do in [50]). We show in Theorem 3 that we can explicitly characterize extremizers of the dual mixed volume inequality and provide visual examples in Figure 4. Finally, we show in the supplemental (Section B.3.1, Theorem 5) that these results can be generalized to a broader class of objectives. These types of challenges were not present in [50]. To our knowledge, this type of result is novel even in the star geometry/Brunn-Minkowski literature, let alone the regularizer learning literature.

- **New applications:** We present new results on the application of star body regularizers that are relevant to both the inverse problems and machine learning communities. These include identifying beneficial properties for optimization, such as weak convexity, and providing guidelines for designing neural network architectures to learn these regularizers. Our findings offer valuable insights into the behavior of these regularizers in downstream tasks and practical methods for learning them. These types of results were not present in [50].

- **New visualizations:** All examples we visualize in the paper are different than those in [50]. The only example that is similar is the one shown in Figure 7 in the supplemental. Our experiment is different, however, since we are showing that this star body induces a weakly convex (squared) gauge, which was not explored in [50].

**3. Impact and significance**

We would like to highlight and summarize the impact and significance of our work:
- Our research introduces novel theory and a new framework for understanding critic-based regularizer learning, addressing a significant gap in existing theoretical foundations.
- We tackle several novel technical challenges inherent to the critic-based setting, distinguishing our work from [50].
- This work also demonstrates the applicability and utility of star geometry tools for regularizer learning and broader machine learning contexts.

---

### Author Response · Authors · 2024-08-12
**New Experimental Results**

Dear reviewers,

We sincerely thank you for your time spent reviewing our work and responding to our rebuttal. We wanted to update you all to let you know that we have conducted experiments based on the comments from some reviewers. In particular, some reviewers expressed concern regarding our lack of empirical results to support our theory. To address these concerns, we have conducted two experiments: one on using the new $\alpha$-divergence loss to learn regularizers and the other on the requirement of positive homogeneity in the neural network layers.

**Denoising comparison of Hellinger and Adversarial Regularizers:** We wanted to test the performance of regularizers learned using the Hellinger-based loss of Eq (5) and those learned using the adversarial loss in Theorem 2.4. To do this, we consider denoising on the MNIST dataset. We take $1000$ random samples from the MNIST training set (constituting our $\\mathcal{D}_r$ distribution) and add Gaussian noise with variance $\\sigma^2 = 0.05$ (constituting our $\\mathcal{D}_n$ distribution). We then aim to reconstruct *test* samples from the MNIST dataset corrupted with Gaussian noise of the same variance seen during training. For our regularizers, we parameterized them with a deep convolutional neural network, similar to the construction in Lunz et al [51]. Specifically, the network has 8 convolutional layers with LeakyReLU activation functions and an additional 3-layer MLP with LeakyReLU activations and no bias terms. The final layer is the Euclidean $\\ell_2$ norm. This network implements a star-shaped regularizer, as outlined in the discussion of our Theorem 4.3. The regularizers were trained using the adversarial loss and Hellinger-based loss (Eq (5)). We also used the gradient penalty term from Lunz et al [51] for both losses.

After training, we then aim to reconstruct noisy *test* samples $y = x_0 + z$ where $x_0$ is a test digit and $z \sim \mathcal{N}(0,\sigma^2I)$. We do this by solving the following with gradient descent initialized at $y$:

$$\\min_{x} ||x - y||^2 + \\lambda \mathcal{R}(x).$$

Here, $\\mathcal{R}$ denotes our trained regularizer. For the adversarially trained network, we used $\lambda := 2 \cdot \tilde{\lambda}$ where $\tilde{\lambda} = \mathbb{E}_{z\sim\mathcal{N}(0,\sigma^2I)}||z||_2$, as described in [51]. For the Hellinger-based network, we found that $\\lambda = 10 \tilde{\lambda}^2$ gave better performance, so we used this for recovery.

We display the average PSNR and MSE over 20 test images for both networks below. We see that the Hellinger-based loss gives competitive performance as compared to the adversarially trained network. This suggests that these new $\alpha$-divergence based losses are potentially worth exploring from a practical perspective as well.

**Noisy image MSE, PSNR:** 0.0486, 13.13

**Hellinger MSE, PSNR:** 0.0046, 23.52

**Adversarial MSE, PSNR:** 0.005, 23.14

**The role of the activation function:** While our theory for neural networks is mainly limited to activation functions such as ReLU and LeakyReLU due to their positive homogeneity, we wanted to understand whether this was a limitation or beneficial from an empirical perspective. Hence we analyzed the influence of the activation function in the above network's performance in denoising. For this study, we changed all LeakyReLU activations in the network to be either the Exponential Linear Unit (ELU), the Gaussian Error Linear Unit (GELU), or the Tanh activation. We trained these networks using the adversarial loss as previously described and then used them for denoising on the same images from the previous experiment. As compared to the network with LeakyReLU activations, we see significant performance degradations when switching to these non-positively homogenous activations. This suggests that potentially positive homogeneity of the activations is a useful property in terms of performance of the neural network-based regularizer.

**LeakyReLU MSE, PSNR:** 0.005, 23.14

**ELU MSE, PSNR:** 0.042, 14.16

**GELU MSE, PSNR:** 0.031, 15.29

**Tanh MSE, PSNR:** 0.049, 13.14

We sincerely thank the reviewers again for their engagement and for raising these concerns. We believe that these additional experiments are valuable in providing support for our theory and will significantly improve the paper quality. We will be sure to include these in the updated version of manuscript.

Please let us know if you have any additional questions or concerns.

---

> ### Comment · Reviewer_kSE2 · 2024-08-13
>
> I would like to thank the authors for the even more extensive numerical comparison. While I would not consider MNIST to be a good example dataset, it is nonetheless more informative than currently provided experiments. I would like to emphasise two particular points which are a bit problematic in the comparison above. In my understanding, in [51] $\lambda$ can be chosen in the form provided, as the regulariser is assumed to be 1-lipshitz. However, by turning to star-bodies I do not believe you have that property, right? This certainly becomes more problematic when considering Hellinger based distances, as you have seen in practice - you need a very different regularisation parameter. As such, above seems like a very fine-tuned comparison, not actually representative of performance, as the Adversarial MSE can likely be fine-tuned to be even better performing by tuning that hyperparameter. I would also suggest including a TV or l1 regulariser to show comparison to model-based regularisation as some sort of baseline.
>
> All in all however, I find this very encouraging, and if the authors agree with the points above and are happy to add these experiments to the paper with a discussion of the points above regarding parameter choice, I would be happy to increase my score.

---

> > ### Author Response · Authors · 2024-08-13
> > **Response to Reviewer kSE2**
> >
> > We thank the reviewer for appreciating the new experimental results and for bringing up these interesting points. Below we further delve into the two main points:
> >
> > **Lipschitzness and regularization strength:** The reviewer is correct that, in general, star body/star-shaped gauges may not be 1-Lipschitz. There are certain conditions that can guarantee Lipschitzness. For example, in the paper, we mention a result that for a star body $K$, the map $x \mapsto ||x||_K$ is 1-Lipschitz if and only if the kernel of $K$ contains the unit Euclidean ball $B^d \subseteq \mathrm{ker}(K)$. When $K$ is star-shaped and not a star body, this condition no longer becomes necessary and sufficient, but there are other ways one can achieve Lipschitzness. For example, when the star-shaped regularizer is parameterized by a deep neural network with positively homogenous activations and linear layers (convolutional or fully connected with no bias terms), it is possible for the regularizer to be 1-Lipschitz by either enforcing each activation to be 1-Lipschitz with linear layers of spectral norm at most 1 or enforcing unit gradient norm, as done in [51].
> >
> > We agree with the reviewer's intuition that the Hellinger-based loss potentially learns a network that is less Lipschitz than the one learned via Adversarial regularization, hence resulting in the need for a different choice of regularization strength when used for inverse problems. In addition to the reviewer's point, we hypothesize this may also result from the fact that the Hellinger loss and adversarial loss weight the distributions $\mathcal{D}_r$ and $\mathcal{D}_n$ differently, resulting in different regularity properties of the learned regularizer. We mainly fixed the Adversarial regularizer's $\lambda$ to the reported value because this was the value recommended in [51], but we will be sure to perform further testing on $\lambda$ and add a more in-depth discussion regarding how these choices relate to Lipschitzness.
> >
> > **Including baselines:** Thank you for this suggestion. We went with the reviewer's suggestion and considered TV regularization. Under the same experimental setup as before, after tuning the regularization strength, we found that TV yielded the following MSE and PSNR:
> >
> > **TV MSE, PSNR:** 0.009, 20.3
> >
> > To summarize, we appreciate and fully agree with the reviewer's suggestions on providing a more in-depth discussion regarding hyperparameter choices. In an updated version of the manuscript, we will add
> > - a discussion regarding parameter choices for both methods when employed to solve inverse problems
> > - updated results with further testing on parameter choices for all methods on more images
> > - TV denoising results as a model-based baseline
> >
> > We thank the reviewer again for their continued support and encouragement. We sincerely appreciate raising your score!

---

### Decision · Program_Chairs · 2024-09-25

**Decision:**

Accept (poster)

**Comment:**

This paper studies the optimization of critic-based loss functions to learn star-shaped gauges regularizers. It uses tools from star geometry and dual Brunn-Minkowski theory which allow deriving the optimal regularizer in certain settings. The paper claims to highlight how "tools from star geometry can aid in understanding the geometry of unsupervised regularizer learning."

The initial reviews for this paper were mixed, with the main concerns being around novelty, usefulness and practical implications of the proposed methodology, and some concerns on the experimental part. However, after the authors' rebuttal, all these concerns have been addressed, the contributions of the paper have been highlighted, and new experiments have been included. As a consequence, now all four reviewers support acceptance of the paper.

I believe this work makes some interesting theoretical contribution, and it additionally sheds light on the practical side through the proposed neural network layers that are compatible with the regularizer. To improve the paper, I strongly encourage the authors to incorporate the points discussed with Reviewer kSE2 regarding hyperparameter choices and updated results including TV denoising.